# A-to-I RNA editing of *CYP18A1* mediates transgenerational wing dimorphism in aphids

**Bin Zhu[1], Rui Wei[1], Wenjuan Hua[1], Lu Li[1], Wenlin Zhang[2], Pei Liang[1]\***

[1]Department of Entomology, College of Plant Protection, China Agricultural University, Beijing, China; [2]Berry Genomics Corporation, Beijing, China

## eLife Assessment

This study presents an **important** finding on the molecular mechanism for transduction of environmentally induced polyphenism. The evidence supporting the claims of the author is **solid**. This paper would be of interest to those studying aphids wing dimorphism.

**Abstract** Wing dimorphism is a common phenomenon that plays key roles in the environmental adaptation of aphid; however, the signal transduction in response to environmental cues and the regulation mechanism related to this event remain unknown. Adenosine (A) to inosine (I) RNA editing is a post-transcriptional modification that extends transcriptome variety without altering the genome, playing essential roles in numerous biological and physiological processes. Here, we present a chromosome-level genome assembly of the rose-grain aphid *Metopolophium dirhodum* by using PacBio long HiFi reads and Hi-C technology. The final genome assembly for *M. dirhodum* is 447.8 Mb, with 98.50% of the assembled sequences anchored to nine chromosomes. The contig and scaffold N50 values are 7.82 and 37.54 Mb, respectively. A total of 18,003 protein-coding genes were predicted, of which 92.05% were functionally annotated. In addition, 11,678 A-to-I RNA-editing sites were systematically identified based on this assembled *M. dirhodum* genome, and two synonymous A-to-I RNA-editing sites on *CYP18A1* were closely associated with transgenerational wing dimorphism induced by crowding. One of these A-to-I RNA-editing sites may prevent the binding of miR-3036-5p to *CYP18A1*, thus elevating CYP18A1 expression, decreasing 20E titer, and finally regulating the wing dimorphism of offspring. Meanwhile, crowding can also inhibit miR-3036-5p expression and further increase CYP18A1 abundance, resulting in winged offspring. These findings support that A-to-I RNA editing is a dynamic mechanism in the regulation of transgenerational wing dimorphism in aphids and would advance our understanding of the roles of RNA editing in environmental adaptability and phenotypic plasticity.

**\*For correspondence:**
liangcau@cau.edu.cn

## Introduction

The rose-grain aphid, *Metopolophium dirhodum* (Walker) (Hemiptera: Aphididae), is one of the most common and economically important aphid pests of cereals, including wheat, barley, rye, and oat, worldwide (*Cannon, 1986*; *Honĕk, 1991*; *Ma et al., 2004*; *Li et al., 2020a*). *M. dirhodum* is native in the Holarctic and then introduced into South America, South Africa, Australia, and New Zealand (*Martinkova et al., 2018*; *Blackman and Eastop, 2000*). Under the continental climate of Central Europe, *M. dirhodum* is usually the most abundant aphid species on cereals (*Martinkova et al., 2018*; *Honĕk, 1987*; *Praslicka, 1996*). In China, *M. dirhodum* was first recorded in the 1980s and then gradually spread from the western to eastern side of the wheat-growing regions, resulting in increased

crop yield reduction (*Gong et al., 2021*). *M. dirhodum* damages cereals by sucking the juice from wheat leaves, stems, and young ears, further resulting in the deterioration of plant nutrition (*Holt et al., 1984*). This aphid defecates sticky honeydew that further obstructs photosynthesis and reduces wheat quality (*Jiang et al., 2019*) and transmits a number of pathogenic plant virus, including the barley yellow dwarf virus (*Kennedy and Connery, 2005*). The nymphs and adults of this aphid may cause yield losses of 27–30% during the latter part of flowering stages of wheat (*Holt et al., 1984*; *Sánchez Chopa and Descamps, 2012*).

Wing polymorphism is commonly observed in insects of various orders, including Hemiptera, Coleoptera, Hymenoptera, Orthoptera, Diptera, Lepidoptera, Isoptera, Psocoptera, and Dermaptera (*Zhang et al., 2019a*). Similar to most aphids, *M. dirhodum* can produce wing morphs when exposed to crowding, poor nutrition, and temperature or photoperiod changes (*Müller et al., 2001*; *Braendle et al., 2006*; *Zhang et al., 2019b*). Wing dimorphism in insects is an adaptive switch to environmental changes. In particular, wingless morphs allocate additional resources to reproduction, enabling rapid colony growth. Meanwhile, winged morphs focus on dispersal, which enables them to look for new habitats and food resources. Compared with wingless ones, winged morphs are better at long-distance migration and host alternation, thus causing more serious host damage and virus transmission (*Zhang et al., 2019a*; *Zhang et al., 2019b*). Great breakthrough has been recently achieved in the exploration on the molecular mechanism underlying the wing differentiation of planthopper. Transcription factors, Zfh1 and FoxO, were verified to regulate alternative wing morphs by faithfully relaying the insulin signaling activity in *Nilaparvata lugens*, providing a new dimension to the molecular explanations of this unique feature in insects (*Xu et al., 2015*; *Xu and Zhang, 2017*; *Lin et al., 2016*). Furthermore, miR-34 could regulate wing dimorphism by mediating the cross-talk among insulin signaling pathway, 20-hydroxyecdysone (20E), and juvenile hormone in a positive feedback manner in *N. lugens* (*Ye et al., 2019*).

Aphids and planthoppers may develop two different strategies in wing dimorphism. Planthopper nymphs can respond to environmental cues to develop into long- or short-winged morphs (*Zhang et al., 2019a*). Meanwhile, wing polymorphism in aphids is usually transgenerational. The parthenogenetic mother could produce either winged or wingless offspring in response to different environmental factors (*Brisson et al., 2010*). Hence, different regulation mechanisms may exist between planthoppers and aphids. *Vellichirammal et al., 2017* observed that a relatively high number of wingless offspring were produced after treatment with 20E or its analogue. Conversely, many winged offspring were produced when ecdysone signaling was suppressed by RNA interference (RNAi) targeting the ecdysone receptor (EcR) (*Vellichirammal et al., 2017*). A hypothesized model suggested that the elevated maternal ecdysone signaling may suppress the embryonic insulin/insulin-like growth factor (IIS) signaling and result in the high expression of FoxO-targeted genes for producing wingless offspring; the opposite is true for producing winged offspring (*Grantham et al., 2020*). These results indicated that 20E is an essential regulatory factor underlying transgenerational plasticity in wing-dimorphic aphid.

In RNA editing, the sequence of an RNA is post-transcriptionally altered (*Sapiro et al., 2019*; *Cuddleston et al., 2022*). Adenosine (A) to inosine (I) editing is the most prevalent type of RNA post-transcriptional modification catalyzed by adenosine deaminase (adenosine deaminase acting on RNA [ADAR]) that uses double-stranded RNAs (dsRNAs) as substrate. Owing to the highly similar structures of I and guanosine (G), cellular machineries usually recognize I as G during translation (*Duan et al., 2018*; *Duan et al., 2017*). High-throughput sequencing has greatly facilitated the genome-wide identification of RNA-editing events, and A-to-I RNA-editing sites have been systematically characterized in various organisms, such as humans (*Picardi et al., 2017*), macaques (*An et al., 2019*), mice, (*Licht et al., 2019*), worms (*Rajendren et al., 2021*), and cephalopods (*Liscovitch-Brauer et al., 2017*). In this work, we generated a high-quality reference genome of *M. dirhodum* and performed transcriptome analyses to systematically identify the A-to-I RNA-editing sites of all transcripts.

The A-to-I mRNA-editing events of numerous genes in 20E biosynthesis and signaling were examined, and two synonymous A-to-I RNA-editing sites on *CYP18A1* were identified to be closely associated with transgenerational wing dimorphism induced by crowding. CYP18A1 plays key roles in 20E metabolism, and its abnormal expression could significantly change the 20E titer, resulting in developmental disorders in multiple insect species (*Guittard et al., 2011*). Furthermore, we identified that one synonymous A-to-I mRNA-editing site may change the binding ability of miR-3036-5p to

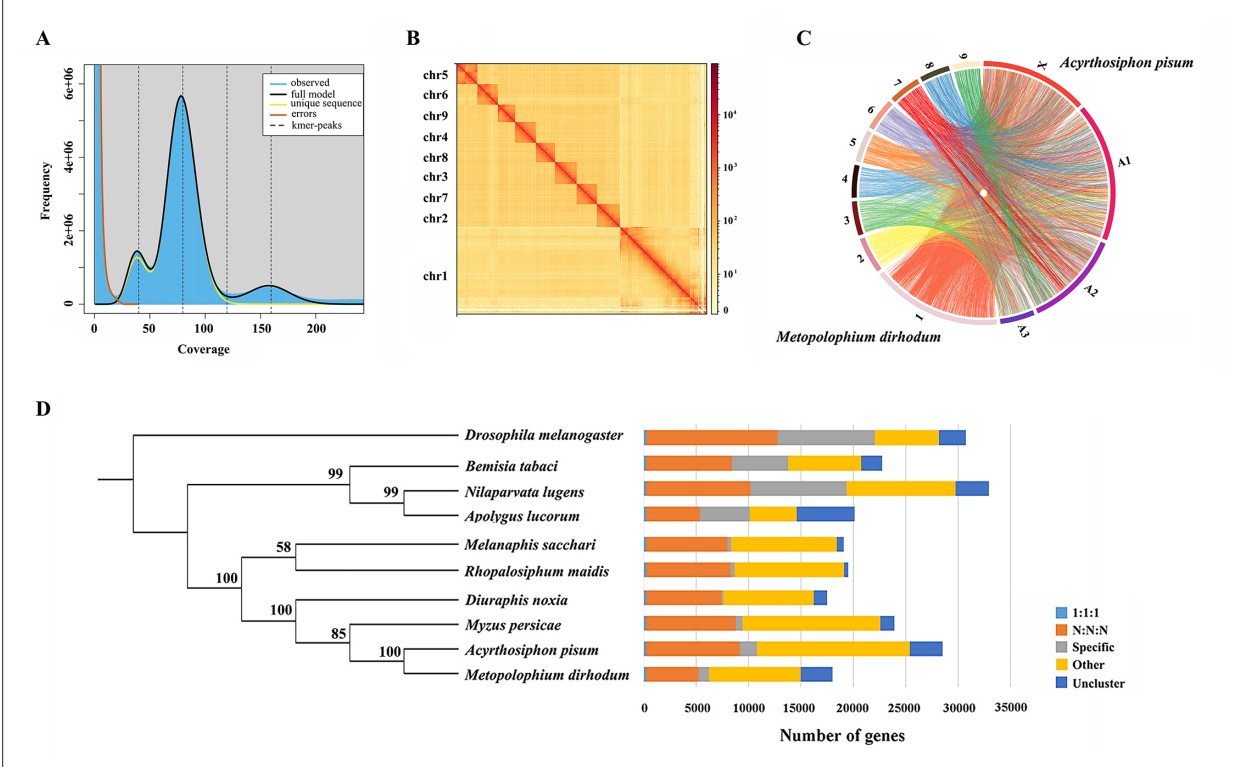

**Figure 1.** Assembled genome for *Metopolophium dirhodum*. (**A**) k-mer (K=17) distribution of Illumina genome sequencing reads. (**B**) Hi-C contact heat map of the assembled genome. (**C**) Chromosome-level synteny analysis between *M. dirhodum* and *Acyrthosiphon pisum*. (**D**) Maximum likelihood phylogeny of *M. dirhodum* and nine other insect species based on a concatenated alignment of the conserved single copy orthologues. The histograms are subdivided to represent different categories of orthology: 1:1:1 (single copy orthologous genes in communal gene families); N:N:N (multiple copy orthologous genes in communal gene families); specific (genes from unique gene families from each species); other (genes that do not belong to any of the above mentioned orthologous categories); uncluster (genes that do not cluster to any families).

The online version of this article includes the following figure supplement(s) for figure 1:

**Figure supplement 1.** Venn diagram of functional annotation based on five databases for *Metopolophium dirhodum*.

*CYP18A1*, thus affecting the CYP18A1 expression, 20E metabolism, and wing dimorphism of offspring in *M. dirhodum*. These findings revealed a new crucial regulatory mechanism of phenotypic plasticity in aphids.

## Results

### Chromosomal-level de novo genome of *M. dirhodum*

A total of 41.17 Gb of high-quality paired-end reads were obtained by Illumina genomic sequencing (~92.22× coverage, *Supplementary file 1*). The genome size of *M. dirhodum* was estimated to be 457.2 M based on *k*-mer counting. *k*-mer distribution analysis revealed a peak at 79.8× of the sequencing depth that suggested a moderate level of heterozygosity (0.445%) and highly repetitive sequence content (59.20%) in the genome (*Figure 1A*), similar to those in other Aphidinae insects with low or moderate level of heterozygosity (*Jiang et al., 2019*; *Chen et al., 2019a*). To obtain a reference genome for *M. dirhodum*, we generated 161.53 Gb of PacBio long reads using a CCS model (*Supplementary file 1*) and subsequently corrected them into 10.34 Gb HiFi reads. The genome was initially assembled using hifiasm, resulting in 296 contigs with a contig N50 of 7.82 Mb and the longest contig of 23.64 Mb (*Table 1*). The 41.17 Gb of short reads generated by Illumina NovaSeq 6000 platform were then mapped against our assembly, resulting in a mapping rate of 92.18%. BUSCO analysis showed that 96.9% (single-copied gene: 92.5%; duplicated gene: 4.4%) of the 1367 single-copy genes in the insecta_odb10 database were identified as complete, 0.4% were fragmented, and 2.7% were missing in the assembled genome. The percentage of complete single-copy genes is higher than

**Table 1.** Assembly features for genomes of *Metopolophium dirhodum* and other Aphidinae insects.

| Genome assembly/ species | M. dirhodum | S. graminum | S. miscanthi | R. maidis | A. pisum | E. lanigerum | D. noxia | M. persicae | A. gossypii |
|---|---|---|---|---|---|---|---|---|---|
| Level | Chr. | Chr. | Chr. | Chr. | Chr. | Chr. | Scaf. | Scaf. | Scaf. |
| No. chr. | 9 | 6 | 9 | 4 | 5 | 6 | - | - | - |
| Size (Mb) | 447.8 | 499.2 | 397.9 | 326 | 541.1 | 330 | 393 | 347.3 | 294 |
| No. contig | 296 | 276 | 1148 | 689 | 60,623 | 12,703 | 49,357 | 6044 | 22,569 |
| Contig N50 (bp) | 8,194,998 | 28,074,450 | 1,638,329 | 9,046,396 | 28,192 | 165,675 | 12,578 | 218,922 | 45,572 |
| No. scaf. | 68 | 22 | 656 | 220 | 23,924 | 7929 | 5641 | 4021 | 4724 |
| Scaf. N50 (bp) | 39,359,500 | 104,490,323 | 36,263,045 | 93,298,903 | 518,546 | 4,427,088 | 397,774 | 435,781 | 437,960 |
| No. gene | 18,003 | 13,353 | 16,006 | 17,629 | 36,195 | 28,186 | 19,097 | 23,910 | 14,694 |

those in the genomes of some other insect species, such as *Sitobion miscanthi* (90.2%) (*Jiang et al., 2019*), *Rhopalosiphum maidis* (94.5%) (*Chen et al., 2019a*), *Acyrthosiphon pisum* (93.5%) (*International Aphid Genomics Consortium, 2010*), and *Eriosoma lanigerum* (96.8%) (*Biello et al., 2021*). Given the moderate level of heterozygosity and the high-level repetitiveness of the genome, these results indicate the high-quality genome assembly of *M. dirhodum*.

For the chromosome-level assembly, 38.09 Gb of clean reads (150 bp paired-end) were obtained from the Hi-C library (coverage: 85.31×, *Supplementary file 1*). A total of 118,367,396 (86.83%) reads were mapped to the draft genome. Among these sequences, 96,331,684 (70.67%) were uniquely mapped and then analyzed with 3D-DNA software to assist genomic assembly. Sixty-eight scaffolds were assembled with an N50 length of 37.54 Mb (*Table 1*). Finally, 447.8 Mb genomic sequences (accounting for 98.50% of the whole assembled length) were located on nine chromosomes (*Figure 1B*, *Table 1*, *Supplementary file 2*), which is identical to *S. miscanthi* (*Jiang et al., 2019*). The contig N50 and scaffold N50 of *M. dirhodum* were also higher than those of previously reported aphid genome assemblies (*Table 1*). This is the first high-quality chromosome-level genome of *M. dirhodum*, which will be very helpful for the cloning, functional verification, and evolutionary analysis of genes in this important species or even other Hemiptera insects.

## Genome annotation

RepeatMasker (*Tarailo-Graovac and Chen, 2009*) and RepBase (*Bao et al., 2015*) were used to annotate repeat sequences. In total, 34.97% of the *M. dirhodum* genome was annotated as repeat sequences. Long terminal repeats (LTRs), long interspersed nuclear elements (LINEs), and DNA transposons accounted for 9.23%, 2.25%, and 10.33% of the whole genome, respectively, and 13.16% of repeat sequences were annotated as unclassified. A total of 286 tRNAs were predicted by trnascan-SE. Using infernal, we also identified 51 small nucleolar RNAs (snoRNAs), 586 ribosomal RNAs (rRNAs), 73 small nuclear RNAs (snRNAs), 59 microRNAs (miRNAs), 286 tRNAs, and 639 other types of ncRNAs.

After repeat sequences were masked, 18,003 protein-coding genes with a mean CDS length of 1776 bp were identified from the *M. dirhodum* genome using de novo, homology-, and RNA sequencing-based methods. The number of genes in the *M. dirhodum* genome is comparable with that of several other Aphidinae species, such as *S. miscanthi* with 16,006 protein-coding genes (*Jiang et al., 2019*) and *Diuraphis noxia* with 19,097 protein-coding genes (*Nicholson et al., 2015*), but far less than those of *R. maidis* (*Chen et al., 2019a*), *A. pisum* (*International Aphid Genomics Consortium, 2010*), *Myzus persicae* (*Jiang et al., 2013*), and *E. lanigerum* (*Biello et al., 2021*) with 26,286, 36,195, 23,910, and 28,186 protein-coding genes, respectively (*Table 1*). Functional annotation found that 16,548 (91.92%), 9030 (50.16%), and 12,836 (71.30%) genes had significant hits with the proteins catalogued in NR, SwissProt, and eggNOG, respectively. A total of 9260 (51.44%) and 6254 (34.74%) genes were annotated to GO terms and KEGG pathway, respectively (*Figure 1—figure supplement 1*).

## Genome synteny and phylogeny analysis

Whole-genome-based phylogenetic analysis was performed with eight other hemipteran insect species, namely, *M. persicae* (*Jiang et al., 2013*), *D. noxia* (*Nicholson et al., 2015*), *A. pisum* (*International Aphid Genomics Consortium, 2010*), *R. maidis* (*Chen et al., 2019a*), *Melanaphis sacchari* (GCA_002803265.2), *N. lugens* (*Ma et al., 2021*), *Bemisia tabaci* (*Xie et al., 2017*; *Chen et al., 2019b*), and *Apolygus lucorum* (*Liu et al., 2021*), to gain insights into an evolutionary perspective for *M. dirhodum*. *Drosophila melanogaster* (*Hoskins et al., 2015*) was used as the outgroup. A total of 209,881 genes were assigned to 22,945 orthogroups for the 10 species (*Figure 1D*, *Supplementary file 3*). A phylogenetic tree was constructed using the single-copy orthologous genes (*Supplementary file 3*). *M. dirhodum* and the five other Aphididae insects formed an Aphididae cluster, showing that *M. dirhodum* is close to *A. pisum* but separated from *M. sacchari* and *R. maidis* (*Chen et al., 2019a*). Three other Hemiptera insects, namely, *B. tabaci* (*Xie et al., 2017*; *Chen et al., 2019b*), *N. lugens* (*Ma et al., 2021*), and *A. lucorum* (*Liu et al., 2021*), formed another cluster (*Figure 1D*, *Supplementary file 3*).

The syntenic relationships between *M. dirhodum* and *A. pisum* genome were compared. The results reveal high levels of genome rearrangement between the chromosomes of *M. dirhodum* and *A. pisum* and a number of fission and fusion events (*International Aphid Genomics Consortium, 2010*). Chr1 in *M. dirhodum* shares 81.9% of the syntenic blocks of chr X in *A. pisum* (*Figure 1C*, *Supplementary file 4*), indicating that chr 1 might be the sex chromosome in *M. dirhodum* (*Li et al., 2020b*). In addition, chrA1 in *A. pisum* is mainly syntenic to chr2, chr4, chr5, and chr8 in *M. dirhodum*;

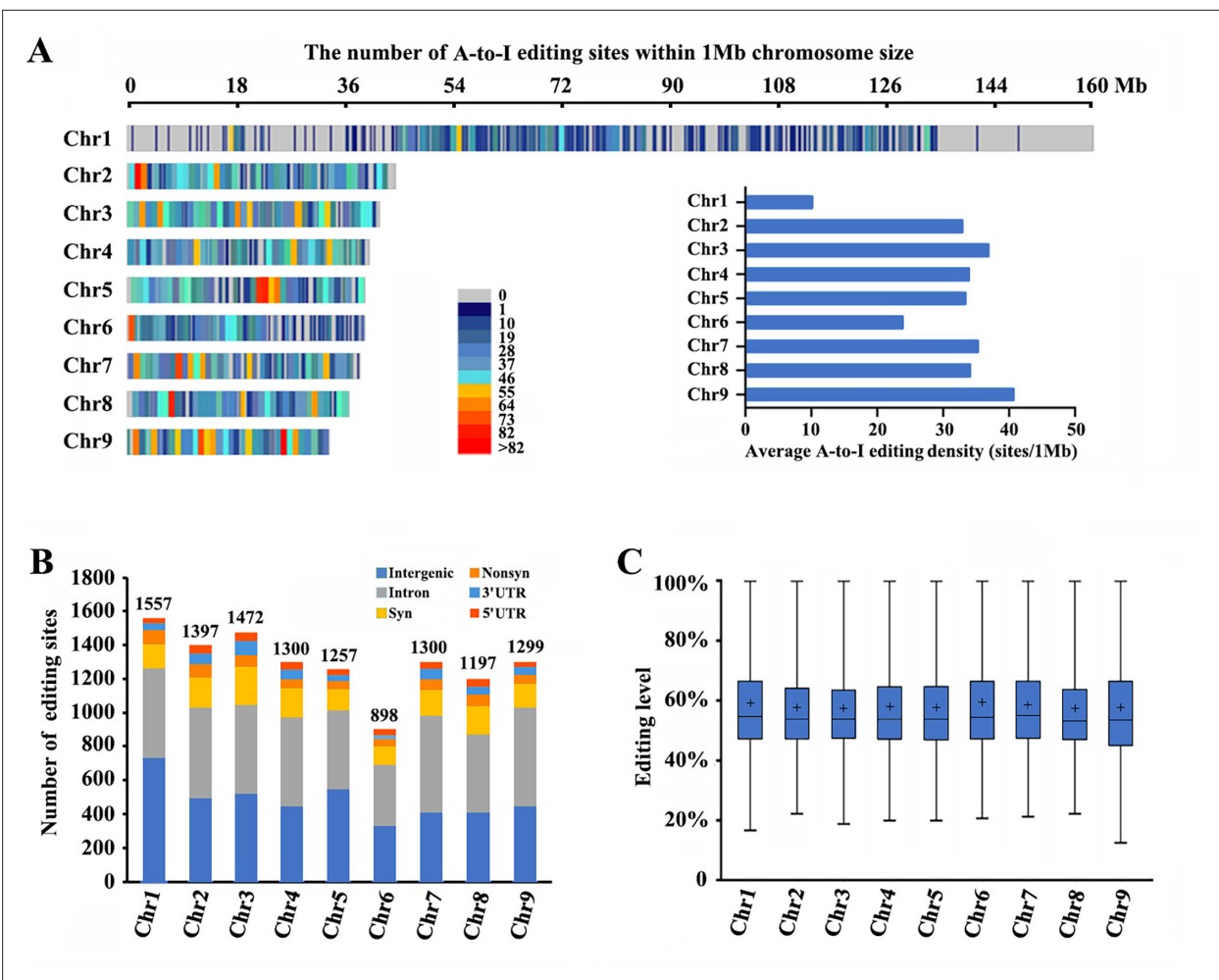

**Figure 2.** Landscape of A-to-I editomes in *Metopolophium dirhodum*. (**A**) Density distribution map of A-to-I RNA-editing sites on nine chromosomes. (**B**) Number and distribution of the detected A-to-I editing sites over different genic regions. (**C**) Average editing levels of the detected A-to-I editing sites on nine chromosomes.

ChrA2 in *A. pisum* is mainly syntenic to chr6, chr7, and chr9 in *M. dirhodum*; and ChrA3 in *A. pisum* is mainly syntenic to chr3 in *M. dirhodum* (*International Aphid Genomics Consortium, 2010*). However, many fusion events covering small regions occurred in all chromosomes between these two insect species (*Figure 1C*, *Supplementary file 4*).

## Identification of A-to-I RNA-editing sites

RNA-seq was performed among winged and wingless *M. dirhodum* third- and fourth-instar nymphs and adults using our assembled genome as a reference to identify potential A-to-I RNA-editing sites in this aphid. After processing, a total of 11,678 A-to-I RNA-editing sites were obtained in *M. dirhodum* (*Supplementary file 5*), of which 1557 are located on chromosome-1, 1397 on chromosome-2, 1472 on chromosome-3, 1300 on chromosome-4, 1257 on chromosome-5, 898 on chromosome-6, 1300 on chromosome-7, 1197 on chromosome-8, 1299 on chromosome-9, and 1 on scaffold 30 (*Figure 2A and B*; *Supplementary file 5*). Among the A-to-I RNA-editing sites, 4356 (37.3%) are in intergenic regions, and 7323 (62.7%) in 3090 protein-coding genes, 342 (2.9%) in 5′ UTRs, 4515 (38.7%) in introns, and 466 (4.0%) in 3′ UTRs; 515 (4.4%) are nonsynonymous (in CDS regions and causes amino acid changes when edited) and 1485 (12.7%) are synonymous (in CDS regions but do not cause amino acid changes) (*Figure 2B*). The average level of these editing sites on chromosome-1 to -9 ranges from 57.2% to 59.3%, showing only a marginal difference (*Figure 2C*). Given the different lengths of the nine assembled chromosomes, the A-to-I RNA-editing density of chromosome-1 (the sex chromosome, chromosome-X) is lower than those of the other eight chromosomes (*Figure 2A*, *Supplementary file 5*).

## A-to-I RNA editing on *CYP18A1* is linked to transgenerational wing dimorphism under crowding

20E is an essential control factor underlying transgenerational plasticity in wing-dimorphic aphids. An environmentally regulated maternal ecdysteroid hormone can mediate wing dimorphism in the next generation (*Vellichirammal et al., 2017*; *Vellichirammal et al., 2016*). The A-to-I RNA editing of numerous genes in 20E biosynthesis and signaling pathway (*Figure 3A*, *Supplementary file 6*; *Luan et al., 2013*), including *CYP306A1*, *CYP302A1*, *CYP315A1*, *CYP18A1*, *ECR*, *FTZ-F1*, and *E74*, was identified. Further verification was performed to confirm whether the A-to-I RNA-editing sites on these genes are involved in wing dimorphism (*Figure 3B*).

Transgenerational wing dimorphism was observed in *M. dirhodum* in which crowding of the parent (100 mother aphids in a 10 cm³ tube) increased the winged offspring (a total of 255 offspring were used to calculate the proportion of winged and wingless individuals in the crowding group, n=255) by 36.4% (*Figure 3E*) compared with that under normal conditions (10 mother aphids in a 10 cm³ tube) (n=277). Nevertheless, 20E treatment for the crowded parent significantly decreased the number of winged offspring (n=272) by 31.2% (*Figure 3E*). The proportions of two synonymous A-to-I RNA-editing sites (editing site 528 and 759, the 528th and 759th nucleotide in the CDS region of *CYP18A1*) (*Figure 3C and D*) on *CYP18A1* were significantly higher in the parents treated with crowding (93.3% and 36.4%) and crowding + 20E (86.6% and 35.6%) than those in the normal group (21.3% and 20.0%) (RNA extraction was individually performed from 150 aphids and then used for reverse transcription and gene amplification in normal, crowding, and crowding + 20E groups) (*Figure 3F*). No evident difference was observed for the A-to-I RNA editing and expression levels of the other genes involved in 20E biosynthesis and signaling pathway after crowding treatment (*Figure 3—figure supplements 1 and 2*, *Supplementary file 6*). Given that CYP18A1 is an essential enzyme in 20E metabolism, these results indicated that the A-to-I RNA editing of *CYP18A1* might be important in crowding induced wing dimorphism in *M. dirhodum*.

The transcriptional level of *CYP18A1* significantly increased by 1.75-fold in the crowded parents compared with that in the normal group, and 20E treatment can further increase its transcription (5.97-fold) (*Figure 3G*). Western blot assay results showed that CYP18A1 expression in crowding (2.90-fold) and crowding + 20E (2.80-fold) treatment groups was also higher than that in the normal group (*Figure 3H*). Moreover, the titer of 20E was significantly decreased in the crowded parent (*Figure 3I*). Thus, we inferred that crowding could affect *CYP18A1* expression and alter its A-to-I RNA-editing level in *M. dirhodum*, thereby controlling the 20E titer in mother aphid to regulate the wing dimorphism of offspring.

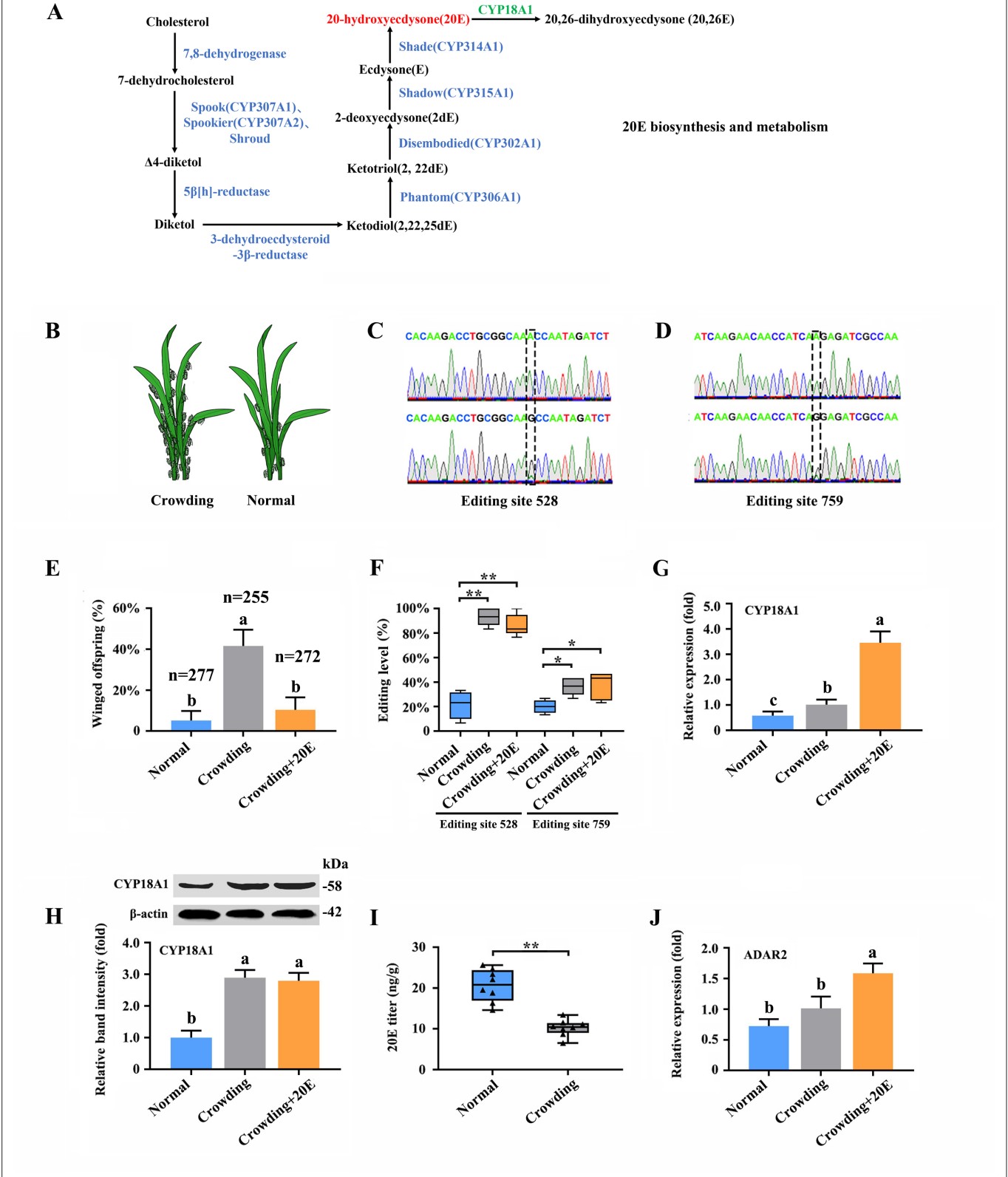

**Figure 3.** A-to-I RNA editing on *CYP18A1* is linked to transgenerational wing dimorphism under crowding condition in *Metopolophium dirhodum*. (**A**) 20-hydroxyecdysone biosynthesis and metabolism pathway. (**B**) Schematic diagram for normal and crowding conditions. (**C, D**) Representative chromatograms of the PCR product direct sequencing for the two synonymous A-to-I RNA-editing sites (editing site 528 and 759, 528th or 759th nucleotide in the CDS region of *CYP18A1*) on *CYP18A1*. (**E**) The proportion of editing individuals under normal, crowding, and crowding + 20E

*Figure 3 continued on next page*

Figure 3 continued

conditions. (**F**) The proportion of winged offspring under normal, crowding, and crowding + 20E conditions. (**G, H**) The expression level of CYP18A1 under normal, crowding and crowding + 20E conditions by RT-qPCR and western blot. (**I**) The 20E titers under normal and crowding conditions. (**J**) The expression level of *ADAR2* under normal, crowding, and crowding + 20E conditions by RT-qPCR. Asterisks indicate significant differences between the treatment and the corresponding control (Student's *t*-test, *0.01<p<0.05, **p<0.01). Different lowercase letters represent significant differences (one-way ANOVA followed by Tukey's multiple comparison tests, p<0.05).

The online version of this article includes the following source data and figure supplement(s) for figure 3:

**Source data 1.** Labeled file for the western blot analysis in *Figure 3H* (CYP18A1).

**Source data 2.** Uncropped file for the western blot analysis in *Figure 3H* (CYP18A1).

**Source data 3.** Labeled file for the western blot analysis in *Figure 3H* (β-actin).

**Source data 4.** Uncropped file for the western blot analysis in *Figure 3H* (β-actin).

**Figure supplement 1.** The proportion of A-to-I RNA-editing individuals for genes involved in 20E biosynthesis and metabolism under normal and crowding conditions.

**Figure supplement 2.** The relative expression for genes involved in 20E biosynthesis and metabolism under normal and crowding conditions.

**Figure supplement 3.** Bioinformatics analysis of the ADAR2 protein.

## RNAi-mediated knockdown of *ADAR2* could regulate the A-to-I RNA-editing level and expression of *CYP18A1*

Insects have completely lost *ADAR1*, and *ADAR2* homologue from mammals may be extremely critical in A-to-I RNA editing (*Figure 3—figure supplement 3*; *Deng et al., 2020*). One *ADAR2* located on chromosome-2 was identified from our genome annotation result. The transcriptional level of *ADAR2* was 2.19-fold higher in the crowding + 20E treatment parent than that in the normal group, but no significant difference was identified between the crowding and normal groups (*Figure 3J*). These results indicated that *ADAR2* expression may not be affected by crowding but can be significantly induced by 20E.

The RNAi-mediated knockdown of *CYP18A1* and *ADAR2* was performed in crowded parent aphids to further explore whether A-to-I RNA editing could regulate CYP18A1 expression and to determine the role of CYP18A1 in transgenerational wing dimorphism. RT-qPCR results showed that compared with that of the control (ds*EGFP*), the relative *CYP18A1* expression decreased by 51.4% and 42.2% (*Figure 4A*) at 48 h post feeding of ds*CYP18A1* and ds*ADAR2*, respectively. Similar results were also obtained by western blot assay (*Figure 4B*). The relative *ADAR2* expression decreased by 46.7% (*Figure 4C*) at 48 h post feeding of ds*ADAR2*. These findings indicated that the RNAi-mediated knockdown of *ADAR2* can affect the abundance of CYP18A1. Moreover, the percentages of A-to-I RNA-editing individuals for the two synonymous editing sites on *CYP18A1* significantly declined by 61.3% and 22.4% (*Figure 4D*) (RNA extraction was individually performed from 150 aphids and then used for reverse transcription and gene amplification in ds*EGFP*, ds*CYP18A1*, and ds*ADAR2* groups), respectively, after *ADAR2* silencing.

The RNAi-mediated knockdown of *CYP18A1* and *ADAR2* can significantly increase the titer of 20E (*Figure 4E*) and reduce the number of winged offspring by 29.6% and 24.4% (*Figure 4F*) (273, 248, and 250 offspring were used to calculate the proportion of winged and wingless individuals among ds*EGFP*, ds*CYP18A1*, and ds*ADAR2* groups, respectively), respectively. All these results showed that *ADAR2* can regulate the A-to-I RNA-editing level of *CYP18A1* to affect its expression and subsequently control the maternal 20E titer, finally influencing the proportion of winged offspring.

## miR-3036-5p targets on *CYP18A1* in *M. dirhodum*

The RNA editing of miRNA binding sites within target transcripts could alter miRNA targeting (*Park et al., 2021*), that is, a miRNA that preferentially targets the edited or unedited version of the transcript and then participates in editing-specific miRNA-mediated transcript degradation. Here, two miRNA-target prediction software programs, miRanda and RNAhybrid, were used to identify the miRNAs that potentially act on *CYP18A1*. The results showed that miR-3036-5p could bind to the sequence containing edited position (editing site 528) of *CYP18A1* in *M. dirhodum* (*Figure 5A*).

MiR-3036-5p was initially identified in *A. pisum*. Meanwhile, the precursor and mature sequences of miR-3036-5p, which was considered a conserved miRNA in aphids, can be excellent matches to

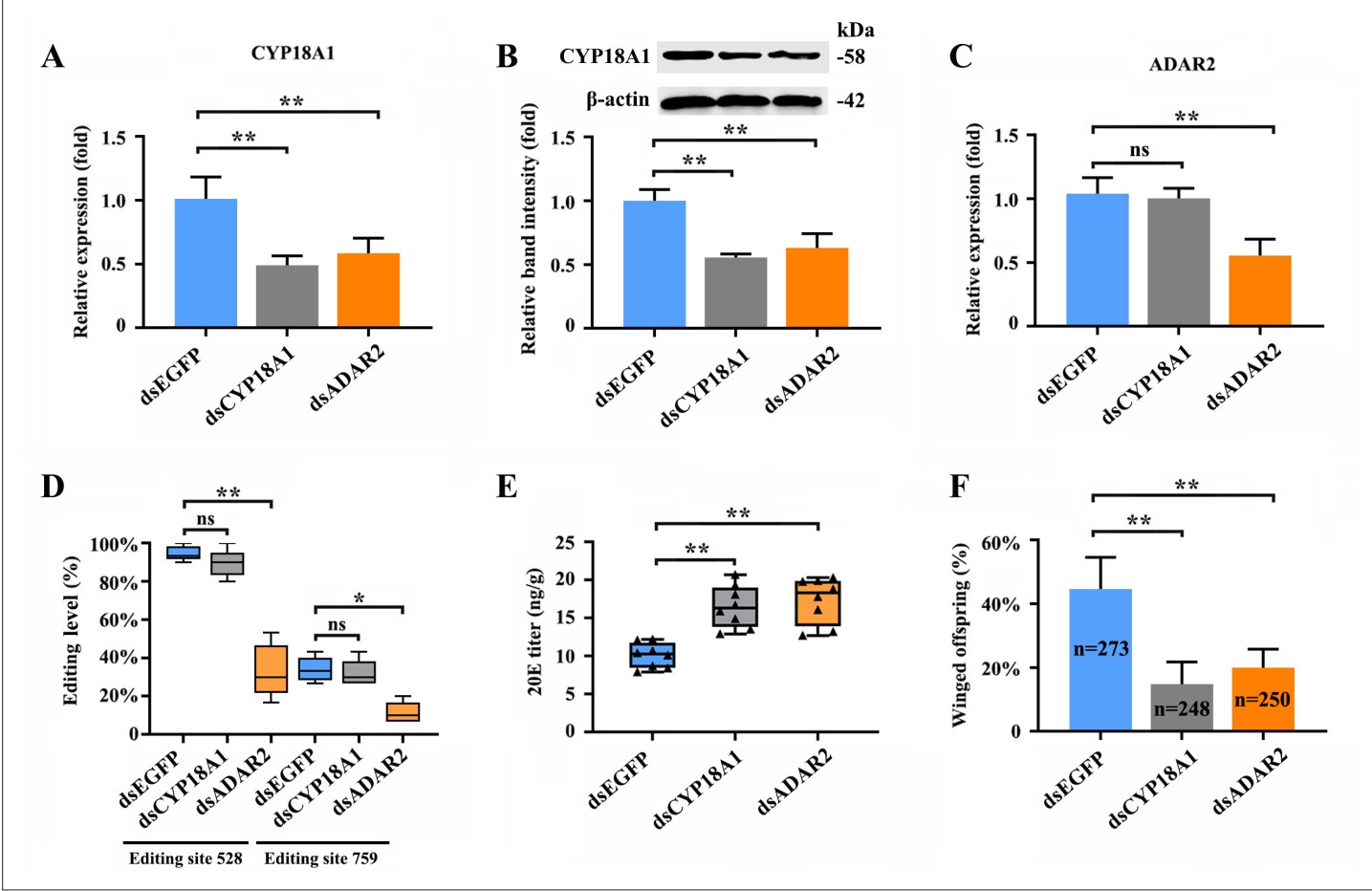

**Figure 4.** RNAi-mediated knockdown of *ADAR2* could regulate the A-to-I RNA-editing level and expression of CYP18A1. (**A, B**) The expression level of CYP18A1 under crowding condition treated with ds*CYP18A1* and ds*ADAR2* after 48 h by RT-qPCR and western blot. (**C**) The expression level of *ADAR2* under crowding condition treated with ds*CYP18A1* and ds*ADAR2* after 48 h by RT-qPCR. (**D**) The proportion of editing individuals under crowding condition treated with ds*CYP18A1* and ds*ADAR2* after 48 h. (**E**) The 20E titers under crowding condition treated with ds*CYP18A1* and ds*ADAR2* after 48 h. (**F**) The proportion of winged offspring under crowding condition treated with ds*CYP18A1* and ds*ADAR2* after 48 h. Asterisks indicate significant differences between the treatment and the corresponding control (Student's *t*-test, *0.01<<0.05, **<0.01).

The online version of this article includes the following source data for figure 4:

**Source data 1.** Labeled file for the western blot analysis in *Figure 4B* (CYP18A1).

**Source data 2.** Uncropped file for the western blot analysis in *Figure 4B* (CYP18A1).

**Source data 3.** Labeled file for the western blot analysis in *Figure 4B* (β-actin).

**Source data 4.** Uncropped file for the western blot analysis in *Figure 4B* (β-actin).

the genome of multiple aphids (*Li et al., 2016*; *Legeai et al., 2010*). MiR-3036-5p expression significantly decreased by 48.8% (*Figure 5B*) after crowding, indicating the potential role of miR-3036-5p in crowding-induced transgenerational wing dimorphism.

RNA immunoprecipitation (RIP) assay using a monoclonal antibody against the Ago1 protein was initially performed to validate the binding between miR-3036-5p and *CYP18A1*. *CYP18A1* was significantly enriched in the Ago1-immunoprecipitated RNAs treated by miR-3036-5p agomir compared with those treated by agomir-NC (*Figure 5C*). Moreover, the colocalization signals of *CYP18A1* and miR-3036-5p were detected by fluorescence in situ hybridization (FISH) assay. *CYP18A1* and miR-3036-5p colocalized in multiple tissues of the whole aphid body, and no signal was observed in the negative control (*Figure 5*, *Figure 5—figure supplement 1*).

We performed reporter assays using luciferase constructs fused to the binding region of *CYP18A1* to further confirm the interaction between *CYP18A1* and miR-3036-5p in vitro. When miR-3036-5p agomir was cotransfected with the pmir-GLO-*CYP18A1*-wt (inserted sequence containing the unedited

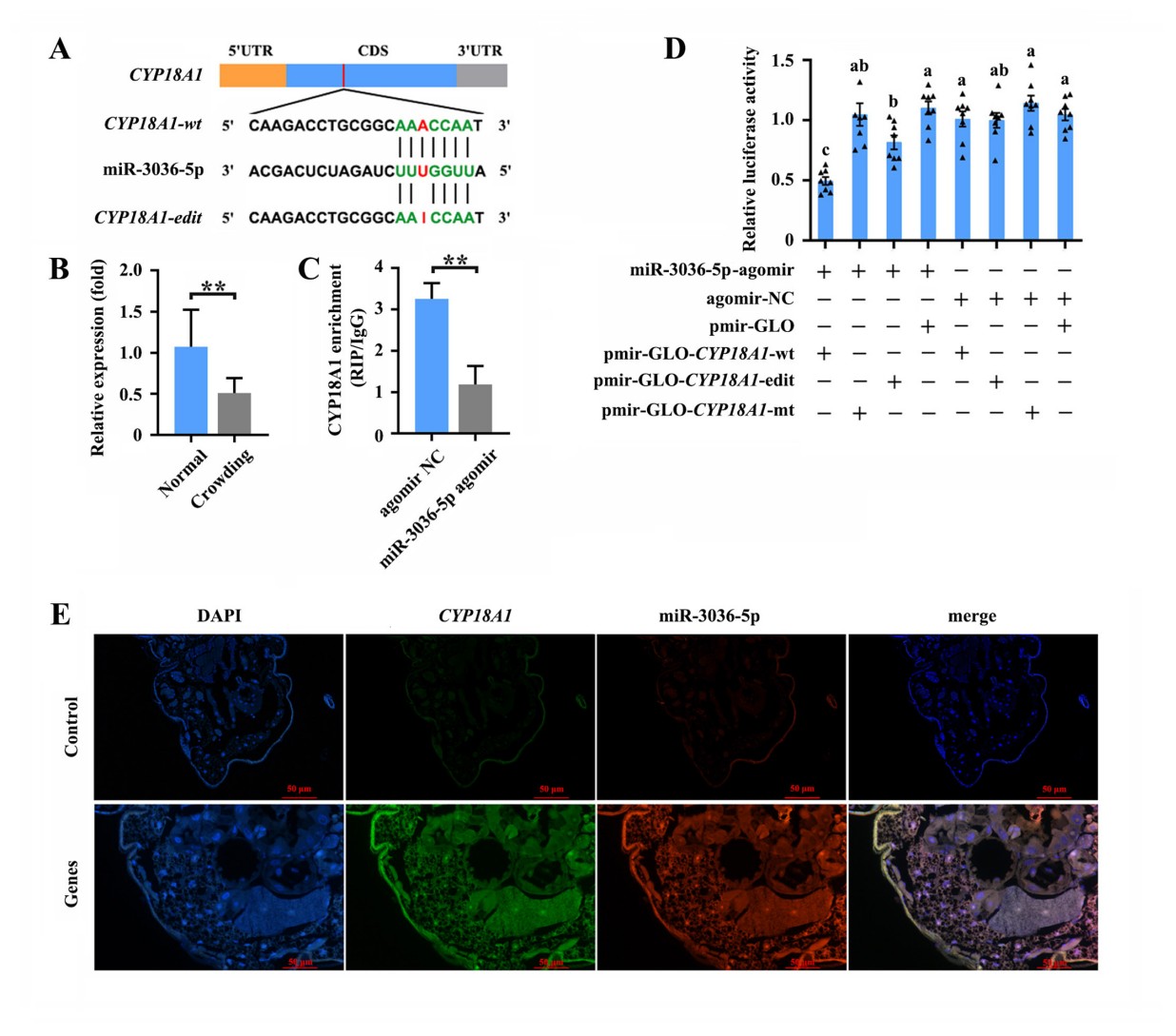

**Figure 5.** miR-3036-5p targets on *CYP18A1* in *Metopolophium dirhodum*. (**A**) The putative miR-3036-5p binding sites in *CYP18A1*. (**B**) The expression level of miR-3036-5p under normal and crowding conditions. (**C**) Interactions between miR-3036-5p and *CYP18A1* determined by RNA-binding protein immunoprecipitation (RIP) in vivo. (**D**) Dual-luciferase reporter assays through co-transfection of miR-3036-5p agomir with recombinant pmirGLO vectors containing wild-type (wt), edited (edit) or mutated (mt) binding sites. (**E**) Co-localization of miR-3036-5p and *CYP18A1* by fluorescence in situ hybridization (FISH) assay. Asterisks indicate significant differences between the treatment and the corresponding control (Student's *t*-test, *$0.01<p<0.05$, **$p<0.01$). Different lowercase letters represent significant differences (one-way ANOVA followed by Tukey's multiple comparison tests, $p<0.05$).

The online version of this article includes the following figure supplement(s) for figure 5:

**Figure supplement 1.** Co-localization of miR-3036-5p and *CYP18A1* by fluorescence in situ hybridization (FISH) assay.

site) into HEK293T cells, the luciferase activity significantly declined by 52.7% (*Figure 5D*) compared with that of the mutated control (pmir-GLO & GSTu1-long-3'UTR-mt, the binding sites complementary to the 'seed' sequences of miR-3036-5p were complete mutated). When miR-3036-5p agomir was cotransfected with the pmir-GLO-*CYP18A1*-edit (inserted sequence containing the edited site) into HEK293T cells, the luciferase activity was not significantly changed relative to that of the mutated control but significantly declined compared with that of the empty vector control (cotransfected with the pmir-GLO) (*Figure 5D*). The luciferase activity in miR-3036-5p agomir and pmir-GLO-*CYP18A1*-wt group significantly declined by 39.2% (*Figure 5D*) compared with that in the miR-3036-5p agomir and pmir-GLO-*CYP18A1*-edit group, indicating the potential influence of A-to-I RNA editing to the binding between miR-3036-5p and *CYP18A1*.

## miR-3036-5p regulates transgenerational wing dimorphism by targeting *CYP18A1* in *M. dirhodum*

To determine the role of miR-3036-5p and its target on transgenerational wing dimorphism in *M. dirhodum*, we combined different treatments by feeding mother aphids with miR-3036-5p agomir and antagomir.

When miR-3036-5p was inhibited at 48 h post feeding of miR-3036-5p antagomir (*Figure 6B*) under normal condition (*Figure 6A*), the transcriptional and translational levels of CYP18A1 were significantly upregulated by 2.89- and 2.24-fold (*Figure 6C and D*), respectively. Meanwhile, the 20E titer was declined by 31.0% (*Figure 6E*), and the proportion of winged offspring increased (n=249 in the antagomir-NC group and n=267 in the miR-3036-5p-antagomir group) by 19.2% (*Figure 6F*). Although CYP18A1 expression was significantly reduced (*Figure 6C and D*) and the 20E titer was increased (*Figure 6E*), the proportion of winged offspring did not change (*Figure 6F*) when treated with miR-3036-5p agomir (*Figure 6B*). These results showed that miR-3036-5p inhibition could induce winged offspring reproduction by regulating CYP18A1 expression, but its overexpression could not play an effective role possibly due to the extremely low proportion of winged offspring (2.80%) under normal growth condition.

Under crowding condition (*Figure 6G*), miR-3036-5p agomir and antagomir treatments (*Figure 6H*) had minimal effects on CYP18A1 expression (*Figure 6I and J*) and could not bring about further changes in the 20E titer (*Figure 6K*) and proportion of winged offspring (*Figure 6L*). Given the high A-to-I RNA-editing level on *CYP18A1* after crowding treatment, the binding between *CYP18A1* and miR-3036-5p might be destroyed. As a consequence, the inhibition or overexpression of miR-3036-5p could not effectively regulate the CYP18A1 expression and wing dimorphism of offspring.

## Discussion

Aphids belong to the superfamily Aphidoidea, which is part of the insect order Hemiptera. More than 5000 species of aphid have been described, and approximately 100 of them are important agricultural pests (*Niu et al., 2019*). Aphids have gradually become important models to study symbiosis, insect–plant interactions, and developmental polyphenism (*Perreau et al., 2021*). Owing to the remarkable advances in sequencing technologies, the genomes of over 20 aphid species have been assembled (*Wei et al., 2022*). In this research, a high-quality chromosome-level genome of *M. dirhodum* was first produced using PacBio long HiFi reads and Hi-C technology. A total of 447.8 Mb genomic sequences were located on nine chromosomes, and 18,003 protein-coding genes were annotated. These genomic resources developed for *M. dirhodum* are valuable for understanding its genetics, development, and evolution and provide important references for the study of other insect genomes.

A-to-I RNA editing is one of the most prevalent forms of post-transcriptional modification in animals, plants, and other organisms (*He et al., 2007*). It acts through multiple mechanisms, including the alteration of protein-coding capacity, generation of diverse protein isoforms, influence on the miRNA binding ability on RNA targets, and alteration of mRNA recognition by RNA binding proteins (*Teoh et al., 2021*). A-to-I RNA editing also plays an important role in normal physiological processes, such as body development (*Buchumenski et al., 2021*), reproduction (*Liu et al., 2017*), and responses to environmental changes (*Rieder et al., 2015*). However, research on A-to-I RNA editing in insect, especially in agricultural pest, is limited. In the present study, 11,678 A-to-I RNA-editing sites were systematically identified in *M. dirhodum* by using the present assembly genome. This work is also the first systematic identification of RNA-editing events in aphids.

Wing polymorphism is commonly observed in insects, resulting from variation in both genetic factors and environmental factors (*Zhang et al., 2019a*). For many aphids, polymorphism is transgenerational (*Vellichirammal et al., 2017*), that is, the mother senses the environment, and her offspring responds with or without wings. High population density is a key environmental factor that induces winged morphs in aphid species (*Vellichirammal et al., 2017*; *Shang et al., 2020*). Here, we also found that a large number of winged offspring of *M. dirhodum* would be generated under crowding conditions. CYP18A1, a key enzyme of 20E inactivation, is involved in crowding-mediated wing dimorphism. CYP18A1 is a cytochrome P450 enzyme with 26-hydroxylase activity, a prominent step for ecdysteroid catabolism in *D. melanogaster*. When CYP18A1 was transfected in *Drosophila* S2 cells, 20E was extensively converted into 20-hydroxyecdysonoic acid (*Guittard et al., 2011*). In *Bombyx*

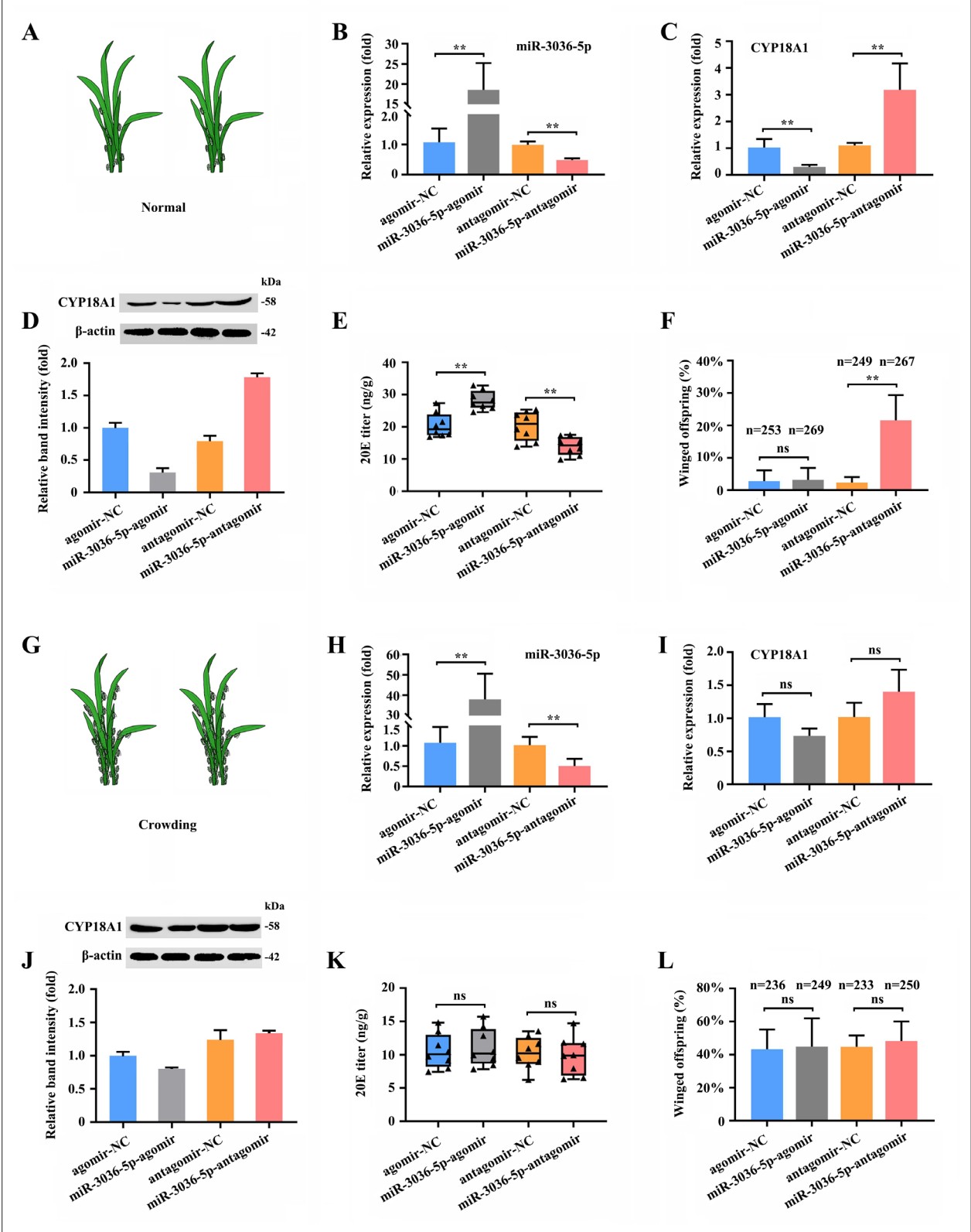

**Figure 6.** miR-3036-5p regulates transgenerational wing dimorphism by targeting *CYP18A1* in *Metopolophium dirhodum*. (**A**) Schematic diagram for normal condition. (**B**) The expression level of miR-3036-5p under normal condition treated with miR-3036-5p agomir or antagomir after 48 h. (**C,** **D**) The expression level of CYP18A1 under normal condition treated with miR-3036-5p agomir or antagomir after 48 h by RT-qPCR (**C**) and western blot (**D**). (**E**) The 20E titers under normal condition treated with miR-3036-5p agomir or antagomir after 48 h. (**F**) The proportion of winged offspring under

*Figure 6 continued on next page*

*Figure 6 continued*

normal condition treated with miR-3036-5p agomir or antagomir after 48 h. (**G**) Schematic diagram for crowding condition. (**H**) The expression level of miR-3036-5p under crowding condition treated with miR-3036-5p agomir or antagomir after 48 h. (**I, J**) The expression level of CYP18A1 under crowding condition treated with miR-3036-5p agomir or antagomir after 48 h by RT-qPCR (**I**) and western blot (**J**). (**K**) The 20E titers under crowding condition treated with miR-3036-5p agomir or antagomir after 48 h. (**L**) The proportion of winged offspring under crowding condition treated with miR-3036-5p agomir or antagomir after 48 h. Asterisks indicate significant differences between the treatment and the corresponding control (Student's *t*-test, *0.01<p<0.05, **p<0.01).

The online version of this article includes the following source data for figure 6:

**Source data 1.** Labeled file for the western blot analysis in *Figure 6D* (CYP18A1).

**Source data 2.** Uncropped file for the western blot analysis in *Figure 6D* (CYP18A1).

**Source data 3.** Labeled file for the western blot analysis in *Figure 6D* (β-actin).

**Source data 4.** Uncropped file for the western blot analysis in *Figure 6D* (β-actin).

**Source data 5.** Labeled file for the western blot analysis in *Figure 6J* (CYP18A1).

**Source data 6.** Uncropped file for the western blot analysis in *Figure 6J* (CYP18A1).

**Source data 7.** Labeled file for the western blot analysis in *Figure 6J* (β-actin).

**Source data 8.** Uncropped file for the western blot analysis in *Figure 6J* (β-actin).

*mori*, CYP18A1 is a 20E-inducible gene, and its ectopic overexpression in transgenic individuals could induce 20E titer reduction, growth arrestment, and larval lethality (*Li et al., 2014*). Considering the essential role of 20E in the wing dimorphism of aphids, we speculated that CYP18A1 might be a key factor mediating this important process. Two synonymous A-to-I RNA-editing sites on *CYP18A1* could be affected by crowding and thus modify the expression of this gene. A-to-I RNA-editing levels are usually affected by changing environments (*Rieder et al., 2015*; *Garrett and Rosenthal, 2012*). For example, one RNA-editing site in $K^+$ channels from polar octopuses greatly accelerates gating kinetics by destabilizing the open state so this species can adapt to the extremely cold environment (*Garrett and Rosenthal, 2012*).

The A-to-I RNA-editing events within mRNA transcripts have the potential to create or destroy the binding site of miRNAs (*Park et al., 2021*). For example, 54,707 A-to-I RNA-editing sites were identified in Tianzhu white yak, and 202 A-to-I-editing sites altered 23 target genes of 140 miRNAs as determined by miRNA–mRNA interaction analysis (*Zhou et al., 2022*). *Nakano et al., 2017* showed that ADAR1 could positively regulate the expression of dihydrofolate reductase (DHFR), which plays a key role in folate metabolism by editing the miR-25-3p and miR-125a-3p binding sites in the 3'-UTR of DHFR, resulting in enhanced cellular proliferation and resistance to methotrexate. Here, we also found that one conserved miRNA named miR-3036-5p in this aphid can target on *CYP18A1*, and one synonymous A-to-I RNA-editing site (editing site 528) on *CYP18A1* could destroy this binding. Under normal condition, this site tends to maintain a low RNA-editing level, resulting in the alteration of *CYP18A1* expression and wing dimorphism of offspring regulated by miR-3036-5p. However, under crowding condition, this site has an extremely high editing level, so miR-3036-5p could not affect CYP18A1 expression. Therefore, the RNA-editing level of *CYP18A1* changes with the population density of mother aphids to affect the accurate regulation mediated by miR-3036-5p and the wing dimorphism of offspring in *M. dirhodum*. Epigenetic modifications to the genome such as DNA methylation and RNA editing are proposed to be a mechanism to regulate phenotypic plasticity by affecting the changes in transcription and subsequent phenotypes (*Duncan et al., 2022*; *Rangan and Reck-Peterson, 2023*). RNA editing, a widespread epigenetic process, is hypothesized to be an adaptive strategy to generate phenotypic plasticity in cephalopods. Squid could rapidly employ RNA editing in response to changes in ocean temperature, and RNA-editing variants of kinesin generated in cold seawater display enhanced motile properties in single-molecule experiments conducted in the cold (*Rangan and Reck-Peterson, 2023*). Our findings supported that insects could also use RNA editing as a mechanism to adapt to environmental pressure and generate phenotypic plasticity.

In conclusion, a high-quality chromosome-level genome of *M. dirhodum* was first assembled, and A-to-I RNA-editing events were identified to be extensively available in this important agricultural pest. Furthermore, we found that two synonymous A-to-I RNA-editing sites on *CYP18A1* could be induced by population density, which is important in crowding-mediated wing dimorphism. One synonymous A-to-I RNA editing at site 528 could inhibit the binding of miR-3036-5p to *CYP18A1*,

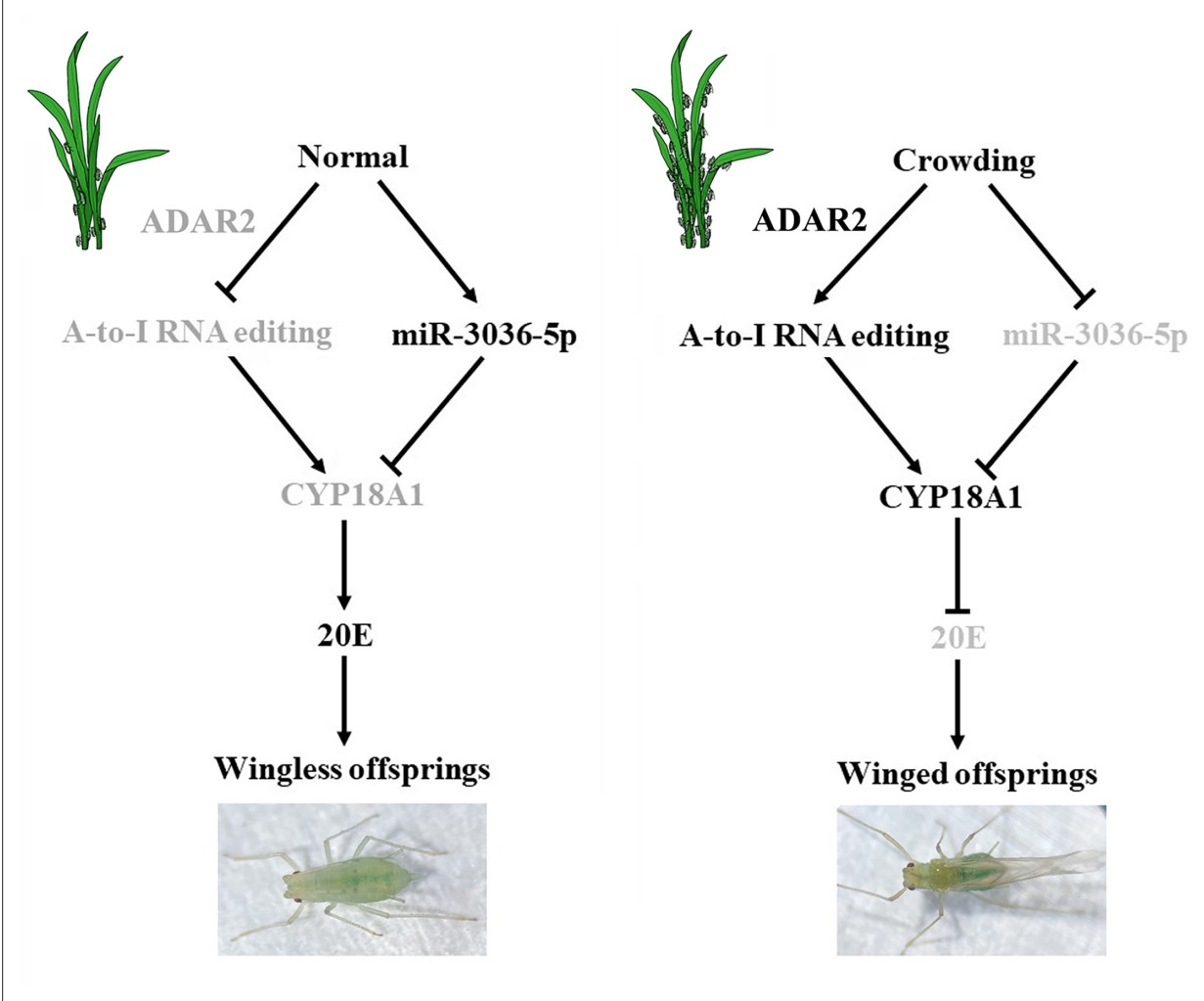

**Figure 7.** Schematic model of the miR-3036-5p-mediated control of transgenerational wing dimorphism by targeting *CYP18A1* in *Metopolophium dirhodum*. The components that are less active or inactive are shown in gray.

thus increasing CYP18A1 expression, decreasing 20E titer, and leading to the wing dimorphism of offspring in *M. dirhodum* (*Figure 7*). Meanwhile, crowding can also inhibit miR-3036-5p expression and further increase CYP18A1 abundance, resulting in winged offspring (*Figure 7*). This work is the first to report that A-to-I RNA editing is involved in insect developmental polyphenism, and the results shine a new light on the functions of RNA editing in phenotypic plasticity in agricultural pests.

## Materials and methods

### Insects

The *M. dirhodum* used in the present study was originally collected from Langfang in Hebei Province, China, in 2018, and then reared on wheat seedlings in our laboratory at 22 ± 2°C and 60% relative humidity with a 16 h light:8 h dark cycle for more than 4 years. All of the progeny were produced by asexual reproduction through parthenogenesis.

### Sample preparation, library construction, and sequencing

Isogenic colonies were started from a single parthenogenetic female of *M. dirhodum* and maintained alone on wheat seedlings prior to the collection of insects for sequencing, respectively. 200 mg of fresh mixed *M. dirhodum*, (including first- to fourth-instar nymphs and winged and wingless adults) was collected for DNA extraction and genome sequencing. Total genomic DNA was extracted using

a Blood & Cell Culture DNA Mini Kit according to the manufacturer's protocol (QIAGEN, Hilden, Germany). For short-read sequencing, a paired-end library (2×150 bp) with short insert sizes of approximately 500 bp was constructed using the VAHTSTM Universal DNA Library Prep Kit for Illumina V2 (Vazyme, Nanning, China) and then sequenced on an Illumina NovaSeq 6000 platform (San Diego, CA). For long-read genomic sequencing, the PacBio SMRTbell 15 kb library was constructed using a SMRTbell Express Template Prep Kit 2.0 (Pacific Biosciences, CA) and then sequenced on the PacBio Sequel II SMRT Cell 8M platform for circular consensus sequencing (CCS) (Pacific Biosciences).

To assist chromosome-level assembly, the Hi-C technique was applied to capture genome-wide chromatin interactions. Approximately 200 mg of fresh *M. dirhodum* with mixed stages (including first- to fourth-instar nymphs and winged and wingless adults) was ground in 2% formaldehyde to allow cross-linking of cellular protein, and approximately 100 µg of DNA was extracted. Subsequently, chromosome integrity and cross-linked protein residues were assessed. Chromatin digestion was performed with the restriction enzyme *Mbo* I. Biotinylated residues were added during repair of the sticky ends, and the resulting blunt-end fragments were ligated under dilute conditions (*Lieberman-Aiden et al., 2009*; *Pan et al., 2021*). The DNA was extracted and randomly sheared to fragments of 300–500 bp. The biotin-labeled fragments were isolated with magnetic beads. The next four steps, including end repair, dA tailing, adapter ligation, and DNA purification, were accomplished by adding the corresponding reaction components sequentially. The library quantity was estimated using Qubit 2.0, an Agilent 2100 instrument (Agilent Technologies, Santa Clara, CA), and quantitative PCR. The Hi-C library was then sequenced using the Illumina NovaSeq 6000 platform with paired-end 150 bp reads.

For PacBio full-length transcriptome sequencing, total RNA was isolated from fresh mixed *M. dirhodum* (including first- to fourth-instar nymphs and winged and wingless adults of equal quality) using an EASYspin Plus Cell/Tissue RNA Isolation Kit (Aidlab Biotechnologies, Beijing, China) and quantified using a NanoDrop ND-2000 spectrophotometer (NanoDrop products, Wilmington, DE). 10 µg of total RNA were reverse transcribed into cDNA using a SMARTer PCR cDNA Synthesis Kit (Takara, Dalian, China) following the manufacturer's protocols. The SMRT library was constructed using the SMRTbell template prep kit (Takara) following the manufacturer's protocols. The library was sequenced on the PacBio Sequel II SMRT Cell 8M platform, and SMRTlink was used to obtain full-length consensus isoform sequences.

For Illumina transcriptome sequencing, total RNA was isolated from winged or wingless *M. dirhodum* of third- and fourth-instar nymphs and adults of equal quality using an EASYspin Plus Cell/Tissue RNA Isolation Kit (Aidlab Biotechnologies) and then quantified using a NanoDrop ND-2000 spectrophotometer. cDNA libraries were constructed using a VAHTSTM mRNA-seq V3 Library Prep Kit (Vazyme, Nanjing, China). A total of 18 libraries were constructed with three biological replicates per sample. Sequencing was performed on an Illumina NovaSeq instrument (Illumina, San Diego, CA), and 150 bp paired-end reads were generated.

## Genome survey and assembly

The K-mer distribution was analyzed to estimate the genome size, heterozygosity, and repeat content using Illumina paired-end reads. The K-mer distribution was analyzed using the Jellyfish and GenomeScope tools based on a k value of 17 (*Vurture et al., 2017*).

PacBio subreads were obtained from the raw polymerase reads after removal of short and low-quality reads and the adaptor sequences, which were then filtered and corrected using the pbccs pipeline with default parameters (https://github.com/PacificBiosciences/ccs; *Hepler et al., 2024*). The resulting HiFi reads (high-fidelity reads) were subjected to hifiasm for de novo assembly (https://github.com/chhylp123/hifiasm; *Cheng et al., 2025*). BWA v0.7.15 (https://sourceforge.net/projects/bio-bwa/; *Li, 2016*; *Li, 2013*) and SAMtools v1.4 (https://sourceforge.net/projects/samtools/; *Whitwham et al., 2017*; *Li et al., 2009*) were used for read alignment and SAM/BAM format conversion. Genome assembly and completeness were assessed using the conserved genes in BUSCO v3.0.2 (https://busco.ezlab.org/; *Simão et al., 2015*).

## Chromosome assembly using Hi-C

The Hi-C sequence data were aligned against the draft genome using JUICER v1.6.2 (https://github.com/aidenlab/juicer; *Durand and Shamim, 2024*; *Durand et al., 2016*). The uniquely mapped

sequences were analyzed using 3D-DNA software (https://github.com/aidenlab/3d-dna; **Dudchenko, 2023**) to assist genomic assembly (**Dudchenko et al., 2017**). The algorithms 'misjoin' and 'scaffolding' were used to remove the misjoins and obtain scaffolds at the chromosomal level. The algorithm 'seal' was employed to find the scaffolds that had been incorrectly removed by the 'misjoin'. The heatmap of chromosome interactions was constructed to visualize the contact intensity among chromosomes using JUICER v1.6.2.

## Genome annotation

Tandem repeats and interspersed repeats were identified using Tandem Repeats Finder (TRF) v4.09 (https://tandem.bu.edu/trf/trf.html; **Benson, 1999**) and RepeatModeler v2.0 (https://github.com/Dfam-consortium/RepeatModeler; **Hubley et al., 2019**; **Flynn et al., 2020**), respectively. Repeat-Masker v4.1.0 (https://www.repeatmasker.org/RepeatMasker/) was used to mask the predicted and known repeated sequences (**Tarailo-Graovac and Chen, 2009**). tRNAscan-SE v1.4alpha (**Chan and Lowe, 2019**) was used to predict tRNAs, and Infernal v1.1.3 (http://eddylab.org/) was used to search the Rfam database v11.0 with an E-value cutoff of $10^{-5}$ to identify other types of noncoding RNAs (ncRNAs) (**Nawrocki and Eddy, 2013**).

Protein-coding genes were predicted through the combination of homology-based, RNA sequencing-based, and ab initio predictions. For the homolog-based approach, the protein sequences of several related species, including *A. pisum* (**International Aphid Genomics Consortium, 2010**), *R. maidis* (**Chen et al., 2019a**), *Diuraphis noxia* (**Nicholson et al., 2015**), *Aphis gossypii* (**Quan et al., 2019**), *Aphis glycines* (**Wenger et al., 2020**), and *Myzus persicae* (**Mathers et al., 2017**), were downloaded from NCBI and aligned against the assembled genome using Gene Model Mapper (GeMoMa) v1.6.1.jar (http://www.jstacs.de/index.php/GeMoMa) to refine the blast hits to define exact intron/exon positions. For the RNA sequencing-based method, the PacBio full-length transcriptome, which was obtained from the pooled sample of *M. dirhodum*, was used to predict the open-reading frames (ORFs) with PASA (https://sourceforge.net/projects/pasa/; **Haas, 2015a**) using default settings. For the ab initio method, two de novo programs, Augustus v3.2.2 (https://bioinf.uni-greifswald.de/augustus/) and SNAP (http://snap.stanford.edu/snap/download.html), were employed with default parameters to predict genes in the repeat-masked genome sequences. All predicted genes from the three approaches were integrated with EVidenceModeler (EVM) (https://sourceforge.net/projects/evidencemodeler/; **Haas, 2015b**) to generate high-confidence gene sets, and the untranslated regions and alternative splicing were predicted using PASA.

The gene set was annotated by aligning protein sequences to functional databases, including NR (nonredundant sequence database), Swiss-Prot, eggNOG (evolutionary genealogy of genes: Nonsupervised Orthologous Groups), GO (Gene Ontology), and KEGG (Kyoto Encyclopedia of Genes and Genomes), using BLAST with a threshold e-value ≤1e-5.

## Phylogeny and comparative genomics

Orthologous groups were identified using the OrthoFinder pipeline (https://github.com/davidemms/OrthoFinder; **Emms and Kelly, 2024**) with default parameters for *M. dirhodum* and nine other species, including *A. pisum* (**International Aphid Genomics Consortium, 2010**), *R. maidis* (**Chen et al., 2019a**), *D. noxia* (**Nicholson et al., 2015**), *M. persicae* (**Mathers et al., 2017**), *M. sacchari* (GCA_002803265.2), *N. lugens* (**Ma et al., 2021**), *B. tabaci* (**Xie et al., 2017**), and *A. lucorum* (**Liu et al., 2021**). *D. melanogaster* (**Hoskins et al., 2015**) was used as an outgroup. MAFFT (https://mafft.cbrc.jp/alignment/software/) was used to align each orthologous gene sequence with default parameters. RAxML was used to infer the maximum-likelihood tree with the best-fit substitution model and 1000 bootstrap replicates. Mummer (https://github.com/mummer4/mummer; **Marçais et al., 2025**) was applied for the detailed collinearity analysis between *M. dirhodum* and *A. pisum* genomes.

## Identification of A-to-I RNA-editing sites

Raw data of FASTQ format (18 libraries constructed from third- and fourth-instar nymphs and adults) were first processed through primary quality control. In this step, clean data (clean reads) were obtained by removing reads containing adapters, reads containing poly-N, low-quality reads (lower than 5), and contaminants from the raw data. Paired-end clean reads were aligned to the assembled genome of *M. dirhodum* using TopHat with default parameters, and all the mapped reads were used

for downstream analyses. The basic principle for identifying an A-to-I RNA-editing site is that the site must be homozygous for gDNA and a mismatch must occur between RNA and DNA. We required that (i) a candidate A-to-I RNA-editing site must be supported by at least three RNA reads that were mapped to overlapping but not identical positions in the reference genome and the site had an editing level >5% (the editing level of a candidate editing site was calculated as the number of reads supporting editing divided by the total number of reads covering that site); (ii) RNA reads with map quality score <30 for the candidate editing positions were discarded; and (iii) candidate sites with multiple editing types were discarded (*Duan et al., 2018*). Given that the aphid species in this study is parthenogenesis and produces haploid progenies, the removal of false positives resulting from heterozygous genomic SNPs is not necessary for subsequent analysis.

## Effect of crowding on winged offspring production

Wingless adults newly emerged within 2 h were collected and separately assigned to two groups: 10 aphids in a 10 ml tube (normal) and 100 aphids in a 10 ml tube (crowding) with wheat seedlings. After treatment for 24 h, the mother aphids were individually moved onto wheat seedlings in a 60 mm diameter Petri dish and allowed to produce nymphs for 12 h. Subsequently, the adult aphids were removed, and the wing phenotypes of offspring were observed after 6–7 days of development.

## dsRNA and miRNA treatment

The dsRNA of *CYP18A1* and *ADAR2* was prepared in vitro using the MEGAscript RNAi kit (Ambion, Foster City, CA). Gene-specific primers containing a T7 polymerase promoter sequence were designed on the E-RNAi website (a tool for the design and evaluation of RNAi reagents for a variety of species, http://www.dkfz.de/signaling/e-rnai3/). The dsRNA of enhanced green fluorescent protein (EGFP) was used as a control. All of the synthesized dsRNAs were dissolved in nuclease-free water and then quantified using a NanoDrop 2000 (Thermo Scientific, Wilmington, DE). MiR-3036-5p mimics (agomir), inhibitors (antagomir), and the respective negative controls (NC agomir, NC antagomir) were synthesized by Shanghai Genepharm Co., Ltd. (Shanghai, China). The subsequent treatment was performed by feeding dsRNA or miRNA agomir/antagomir through wheat seedlings following the method of *Shang et al., 2020* with some modifications (*Shang et al., 2020*). Wheat seedlings were first dried for 12 h and then completely immersed in water solution containing dsRNA (2000 ng/µl) or miRNA mimic/ inhibitor (2 µM) for 1 min. The wheat seedlings were allowed to air dry for 30 min, and their roots were subsequently inserted into a 250 µl PCR tube containing 200 µl of dsRNA (2000 ng/µl) or miRNA mimic/inhibitor (2 µM). The tube was finally transferred to a 100 mm diameter Petri dish. Fourth-instar nymphs were inoculated to these treated wheat seedlings until they reach the adult stage, and the emerged wingless adults were then transferred under normal or crowding condition for continuous dsRNA or miRNA treatment.

## 20-Hydroxyecdysone (20E) treatment and measurement

In brief, 20E (Cayman Chemicals, USA) was dissolved in 95% ethanol as the stock solution, then diluted to 12 mg/ml with ultrapure water, and used as the working solution (*Li et al., 2022*). The wheat seedlings treated with 20E were used to feed the fourth-instar nymphs, and the treatment method was the same as above.

For quantification, 20E was extracted from adults under different treatment conditions and then detected via competitive EIA (Cayman Chemicals, Ann Arbor, MI) using anti-20E rabbit antiserum (Cayman Chemicals). Samples were prepared following the method of *Koyama and Mirth, 2021* (*Koyama and Mirth, 2021*). (1) The collected adults were washed twice in double-distilled water to remove any contaminants. (2) After the adults were dried briefly on a small piece of paper towel for about 1 min, a group of samples for one biological replicate was weighed on an ultramicro balance (20–30 mg is usually sufficient for accurate ecdysone quantification). Eight biological replicates were used in each sample. (3) All weighed samples were placed in a 1.5 ml microcentrifuge tube, added with threefold volume of absolute methanol (30 µl methanol for 10 mg samples), and frozen immediately on dry ice. (4) The frozen samples were homogenized carefully using disposable pestles and a cordless hand-pestle motor and then centrifuged in tubes at 4°C at maximum speed for 5 min. (5) The supernatant was carefully transferred into new 1.5 ml microcentrifuge tubes, and centrifugation was repeated 1–2 times until no precipitate is visible. (6) Methanol was evaporated completely using a

centrifugal vacuum concentrator at room temperature for 1–2 h. Quantification assay was performed in accordance with the manufacturer's instructions.

## MiRNA target prediction and dual-luciferase reporter assay

Two miRNA target prediction software programs, miRanda (http://www.microrna.org) and RNAhybrid (https://bibiserv.cebitec.uni-bielefeld.de/rnahybrid/), were used with default parameters to predict miRNAs that potentially target *CYP18A1*. The miRNAs identified from *A. pisum* were downloaded and used for target prediction analysis. For luciferase activity assay, the luciferase reporter plasmid (PmirGLO vector, Promega, Leiden, the Netherlands) (*Zhu et al., 2021*) was constructed by inserting the wild-type, edited, or mutated target sequences of *CYP18A1* between the firefly luciferase ORF and SV40 poly (A) into the pmirGLO vector (Promega). The constructed vectors, miRNA agomir or NC (negative control) agomir, were transferred into HEK293T cells (HEK293T cells were purchased from ATCC and regularly tested to be Mycoplasma-negative as judged by the absence of extranuclear DAPI staining) using the calcium phosphate cell transfection kit (Beyotime) following the manufacturer's instructions. The activities of firefly and Renilla luciferases were measured at 48 h post transfection with the Dual Glo Luciferase Assay System (Promega). For each transfection, the luciferase activity was averaged from the results of eight replicates.

## Quantitative real-time PCR (qRT-PCR)

Total RNA was extracted using TRIzol reagent (Invitrogen, Carlsbad, CA) following the manufacturer's guidelines. The first strand of complementary DNA was synthesized from 1 µg of total RNA using a PrimeScript RT reagent Kit with gDNA Eraser (Perfect Real Time) (Takara Biotechnology, Dalian, China). The reaction was performed on an ABI 7500 Real Time PCR system (Applied Biosystems, Foster City, CA). The expression levels for each gene were normalized to *β-actin* and *GAPDH* and calculated using the $2^{-\Delta\Delta Ct}$ method (*Livak and Schmittgen, 2001*). Five biological replicates (more than 100 aphids were used for one biological replicate) were used in each sample.

## RNA immunoprecipitation (RIP)

A Magna RIP Kit (Millipore, Germany) was used to perform RIP assay following previous studies. Adults were first fed with miR-3036-5p agomir and then subjected to RIP analysis 24 h later. Approximately 100 aphids were collected and homogenized in ice-cold RIP lysis buffer. The lysates were centrifuged at 13,600 × $g$ for 10 min at 4°C, and the supernatant (100 µl) was incubated with 5 µg of RIP Ab+Ago-1 antibody (Millipore) or normal mouse IgG (Millipore, negative control) beads for 12 h at 4°C. The beads were then washed with RIP wash buffer 2–3 times. Finally, the transcript enrichment ratio for the purified RNAs was determined by qRT-PCR.

## Fluorescence in situ hybridization (FISH)

Antisense nucleic acid detection probes (5′-CAACGAACTAATCACGTTGGTGATGGCGAGACACAG CGAACCGGC-3′) labeled with Cy3 and (5′-TGCTGAGATCTAGAAACCAAT-3′) labeled with FAM (GefanBio, China) were designed to detect CYP18A1 and miR-3036-5p. The random shuffled probe (5′-UUGUACUACACAAAAGUACUG-3′) was used as negative control. For FISH, the adults were fixed in 4% paraformaldehyde for 2 h and then treated with 0.2 M hydrochloric acid and proteinase K. The treated samples were incubated with the mRNA or miRNA probes at 65°C for 48 h in the dark and washed in PBS five times at room temperature. Fluorescence signals were finally analyzed, and images were recorded using a Nikon Eclipse Ci microscope (Tokyo, Japan).

## Western blot

The antibody against CYP18A1 was synthesized by Beijing Protein Innovation Co., Ltd. (Beijing, China), and the antibody against β-actin (TransGen Biotech, Beijing, China) was used as an internal control. Total proteins of each sample were extracted using a Tissue Protein Extraction Kit (Cwbio, Beijing, China), and the concentration was determined using a BCA Protein Assay Kit (Cwbio) following the manufacturer's protocol.

The extracted total proteins were separated by 10% sodium dodecyl sulfate-polyacrylamide gel electrophoresis and then transferred onto polyvinylidene fluoride membranes (Millipore). The membrane was blocked with 5% skim milk (Biotopped, China) for 2 h and subsequently incubated with

specific antibody. Immunoreactivity was imaged with the multifunctional molecular imaging system (Azure). Quantitative analysis of the western blot results was performed using the program ImageJ.

## Statistical analysis

Statistical analysis was performed using GraphPad Prism version 8.0. One-way analysis of variation followed by Tukey's multiple comparisons was used for multiple comparisons ($p < 0.05$), and the Student's *t*-test test (*$p < 0.05$; **$p < 0.01$; ns, no significance) was used for pairwise comparison. All primers used in this study are listed in *Supplementary file 7*.

## Acknowledgements

This work was supported by the China Agriculture Research System (grant number: CARS-05-03A-16). We thank Dr. Yuange Duan (China Agricultural University) for his critical comments on RNA editing analysis, Dr. Yaoguo Qin (China Agricultural University) for her assistance on aphid rearing, Miss Jiajia Song (China Agricultural University) for her drawing, and Berry Genomics Corporation for technical support in Illumina, PacBio, and Hi-C.

## Additional information

### Competing interests

Wenlin Zhang: Employee of Berry Genomics Corporation. The other authors declare that no competing interests exist.

### Funding

| Funder | Grant reference number | Author |
| --- | --- | --- |
| China Agricultural Research System | CARS-05-03A-16 | Pei Liang |

The funders had no role in study design, data collection and interpretation, or the decision to submit the work for publication.

### Author contributions

Bin Zhu, Conceptualization, Resources, Data curation, Formal analysis, Validation, Investigation, Writing - original draft, Writing - review and editing; Rui Wei, Wenjuan Hua, Lu Li, Resources, Validation, Investigation; Wenlin Zhang, Resources, Data curation, Software, Formal analysis; Pei Liang, Conceptualization, Formal analysis, Funding acquisition, Writing - original draft, Project administration, Writing - review and editing

### Author ORCIDs

Bin Zhu https://orcid.org/0000-0002-5632-8019
Pei Liang https://orcid.org/0000-0002-3083-8918

Reviewer #1 (Public Review): https://doi.org/10.7554/eLife.96540.3.sa1
Reviewer #2 (Public Review): https://doi.org/10.7554/eLife.96540.3.sa2
Author response https://doi.org/10.7554/eLife.96540.3.sa3

## Additional files

### Supplementary files

Supplementary file 1. Statistics for sequencing data for *Metopolophium dirhodum* assembly.

Supplementary file 2. Overview of chromosome length of *Metopolophium dirhodum* assembly.

Supplementary file 3. Gene family clusters of 10 species.

Supplementary file 4. Chromosome-level synteny analysis between *Metopolophium dirhodum* and *Acyrthosiphon pisum*.

Supplementary file 5. Overview of the A-to-I RNA-editing sites in *Metopolophium dirhodum*.

Supplementary file 6. A-to-I RNA-editing sites for genes involved in 20E biosynthesis and metabolism pathway.

Supplementary file 7. All primers used in this study.

MDAR checklist

## Data availability

Raw genome sequencing reads and RNA-seq reads were deposited in the National Center for Biotechnology Information using BioProject Accession no. PRJNA751716 and PRJNA751719. The whole genome shotgun sequencing projects have been deposited at NCBI GenBank under accession no. JAIOUA000000000. All data generated or analysed during this study are included in the manuscript and supporting files.

The following datasets were generated:

| Author(s) | Year | Dataset title | Dataset URL | Database and Identifier |
|---|---|---|---|---|
| Zhu B | 2021 | Metopolophium dirhodum isolate:CAU Genome sequencing and assembly | https://www.ncbi.nlm.nih.gov/bioproject/PRJNA751716/ | NCBI BioProject, PRJNA751716 |
| Zhu B | 2021 | Metopolophium dirhodum Genome sequencing and assembly | https://www.ncbi.nlm.nih.gov/bioproject/PRJNA751719/ | NCBI BioProject, PRJNA751719 |
| Zhu B | 2021 | Metopolophium dirhodum isolate CAU, whole genome shotgun sequencing project | https://www.ncbi.nlm.nih.gov/nuccore/JAIOUA000000000 | NCBI GenBank, JAIOUA000000000 |

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
