## [Editor Report · eLife Assessment]

This study presents an **important** finding on the molecular mechanism for transduction of environmentally induced polyphenism. The evidence supporting the claims of the author is **solid**. This paper would be of interest to those studying aphids wing dimorphism.

---

## [Referee Report · Reviewer #1 (Public Review)]

Summary:

In this study, a chromosome-level genome of the rose-grain aphid M. dirhodum was assembled with high quality, and A-to-I RNA-editing sites were systematically identified. The authors then demonstrated that: (1) Wing dimorphism induced by crowding in M. dirhodum is regulated by 20E (ecdysone signaling pathway); (2) an A-to-I RNA editing prevents the binding of miR-3036-5p to CYP18A1 (the enzyme required for 20E degradation), thus elevating CYP18A1 expression, decreasing 20E titer, and finally regulating the wing dimorphism of offspring.

Strengths:

The authors present both genome and A-to-I RNA editing data. An interesting finding is that a A-to-I RNA editing site in CYP18A1 ruin the miRNA binding site of miR-3036-5p. And loss of miR-3036-5p regulation lead to less 20E and winged offspring.

---

## [Referee Report · Reviewer #2 (Public Review)]

Summary:

Environmental influences on development are ubiquitous, affecting many phenotypes in organisms. However molecular genetic and cellular mechanisms transducing environmental signals are still only barely understood. This study examines part of one such intracellular mechanism in a polyphenic (or dimorphic) aphid.

Strengths:

While other published reports have linked phenotypic plasticity to RNA editing before, this study reports such an interaction in insects. The study uses a wide array of molecular tools to identify connections upstream and downstream of the RNA editing to elucidate the regulatory mechanism, which is illuminating.

Weaknesses:

While this system is intriguing, this report does not foster confidence in its conclusions. Many of the analyses seem based on very small sample sizes. It is itself problematic that sample sizes are not obvious in most figures, although based on Methods section covering RNAseq, they seem to be either 3, 6 or 9, depending on whether stages were pooled, but that point is not made clear. With such small sample sizes, statistical tests of any kind are unreliable. Besides the ambiguity on sample sizes, it's unclear what error bars or whiskers show in plots throughout this study. When sample sizes are small estimates of variance are not reliable. Student's t-test is not appropriate for comparisons with such small sample sizes. Presently, it is not possible to replicate the tests shown in Figures 3, 4 and 6. (Besides the HT-seq reads, other data should also be made publicly available, following the journal's recommendations.) Regardless, effect sizes in some comparisons (Fig 3J, 4A-C, 6E,H) are clearly not large, making confidence in conclusions low. The authors should be cautious about over-interpreting these data.

[Editors' note: The authors made a great effort to address the reviewers' concerns. The current manuscript is significantly improved with additional data and clarification.]

---

## [Author Response]

The following is the authors’ response to the original reviews.

**Public Reviews:**

**Reviewer #1 (Public Review):**
Summary:In this study, a chromosome-level genome of the rose-grain aphid M. dirhodum was assembled with high quality, and A-to-I RNA-editing sites were systematically identified. The authors then demonstrated that: (1) Wing dimorphism induced by crowding in M. dirhodum is regulated by 20E (ecdysone signaling pathway); (2) an A-to-I RNA editing prevents the binding of miR-3036-5p to CYP18A1 (the enzyme required for 20E degradation), thus elevating CYP18A1 expression, decreasing 20E titer, and finally regulating the wing dimorphism of offspring.Strengths:he authors present both genome and A-to-I RNA editing data. An interesting finding is that a A-to-I RNA editing site in CYP18A1 ruin the miRNA binding site of miR-3036-5p. And loss of miR-3036-5p regulation lead to less 20E and winged offspring.Weaknesses:How crowding represses the miR-3036-5p is still unclear.
**Reviewer #2 (Public Review):**
Summary:Environmental influences on development are ubiquitous, affecting many phenotypes in organisms. However molecular genetic and cellular mechanisms transducing environmental signals are still only barely understood. This study examines part of one such intracellular mechanism in a polyphenic (or dimorphic) aphid.Strengths:While other published reports have linked phenotypic plasticity to RNA editing before, this study reports such an interaction in insects. The study uses a wide array of molecular tools to identify connections upstream and downstream of the RNA editing to elucidate the regulatory mechanism, which is illuminating.Weaknesses:While this system is intriguing, this report does not foster confidence in its conclusions. Many of the analyses seem based on very small sample sizes. It is itself problematic that sample sizes are not obvious in most figures, although based on Methods section covering RNAseq, they seem to be either 3, 6 or 9, depending on whether stages were pooled, but that point is not made clear. With such small sample sizes, statistical tests of any kind are unreliable. Besides the ambiguity on sample sizes, it's unclear what error bars or whiskers show in plots throughout this study. When sample sizes are small estimates of variance are not reliable. Student's t-test is not appropriate for comparisons with such small sample sizes. Presently, it is not possible to replicate the tests shown in Figures 3, 4 and 6. (Besides the HT-seq reads, other data should also be made publicly available, following the journal's recommendations.) Regardless, effect sizes in some comparisons (Fig 3J, 4A-C, 6E, H) are clearly not large, making confidence in conclusions low. The authors should be cautious about over-interpreting these data.

We appreciate very much for the reviewers’ time spent on our manuscript and the referees for the valuable suggestions and comments.

**To Reviewer #1:**

At present, researches on miRNAs mainly focus on its role in gene regulation by binding to the mRNA of target genes, “how miRNAs are regulated” has received less attention.

Recent researches indicated that the expression of miRNAs is also regulated at the transcriptional or post transcriptional level. Transcriptional regulation including changes in the promoter of microRNA genes, and post-transcriptional mechanisms such as changes in miRNA processing and stability can both affect the final expression level of miRNAs.

This article did not address how crowding treatment regulates miRNA expression. But this will be a very interesting issue, and we will pay attention to it in our future research.

Thank you for this suggestion.

**To Reviewer #2:**

(1) “Transgenerational wing dimorphism was observed in *M. dirhodum* in which crowding of the parent (100 mother aphids in a 10 cm³ tube) increased the winged offspring (Fig 3E).” In this experiment, over 250 offsprings were used to calculate the proportion of winged and wingless individuals in normal (277), crowding (255) and crowding+20E (272) groups, respectively.

“The RNAi-mediated knockdown of *CYP18A1* and *ADAR2* can significantly increase the titer of 20E (Fig. 4E) and reduce the number of winged offspring by 29.6% and 24.4% (Fig. 4F), respectively.” In this experiment, over 245 offsprings were used to calculate the proportion of winged and wingless individuals in ds*EGFP* (273), ds*CYP18A1*(248), and ds*ADAR2* (250) groups, respectively.

“miR-3036-5p agomir and antagomir treatments could affect the proportion of winged offspring under normal conditions (Fig. 6F), but have no effect on the wing dimorphism of offspring under crowded conditions (Fig. 6L).” In this experiment, over 235 offsprings were used to calculate the proportion of winged and wingless individuals in each group, respectively.

So I think our conclusion that crowding treatment, A-to-I RNA editing, and miRNAs could affect the wing dimorphism of offspring in *M. dirhodum* is very reliable. Because the number of aphids we use to count the results is sufficient.

(2) The quantitative PCR method is used to detect changes in gene expression levels of *CYP18A1* and *ADAR2* after treatment with crowding, 20E, dsRNA, miRNA agomir and antagomir, and the results are shown in Fig. 3J, 4A-C, 5B, 6B, H, respectively. 5 biological replicates (more than 100 aphids were used for each biological replicate) were used in each sample, which might be sufficient for qPCR experiments. And among these biological replicates, the differences in gene expression levels are relatively small.

(3) The titer of 20E was detected after treatment with crowding, 20E, dsRNA, miRNA agomir and antagomir, and the results are shown in Fig. 3I, 4E, 6E, K, respectively. 8 biological replicates (more than 100 aphids were used for each biological replicate) were used in each sample.

The number of biological replicates used in each analysis and the number of aphids included in each biological replicate have been added in the Materials and Methods section. Thank you very much for pointing out this important issue.

**Reviewer #1 (Recommendations For The Authors):**
Several questions:(1) This study was conducted on the rose-grain aphid *M. dirhodum*. However, pea aphid Acyrthosiphon pisum seems to be a better object in wing dimorphism and development studies. Have the authors also identified the A-to-I RNA editing on pea aphids or other aphids?

Wheat is one of the main grain crops in China as well as in the world. *Metopolophium dirhodum* is one of the most important wheat aphids around China, and has posed a significant threat to grain production. The current study was conducted to determine the regulatory mechanism of wing dimorphism on *M. dirhodum*, which might be of great significance to better control this pest in wheat production.

Surely the pea aphid offers more established experimental tools and genomic resources. However, with the development of high-throughput sequencing technology, the chromosome level genomes of many insect species have been assembled. That means any of various insects might be studied as a model species, and not limited to *Drosophila melanogaster*, *Acyrthosiphon pisum*, etc.

We didn’t identify the A-to-I RNA editing on pea aphids or other aphids. A recent study has shown that editing events are poorly conserved across different *Xenopus* species. Even sites that are detected in both *X. laevis* and *X. tropicalis* show largely divergent editing levels or developmental profiles. In protein-coding regions, only a small subset of sites that are found mostly in the brain are well conserved between frogs and mammals. The conservation of RNA editing in aphids is still unknown, and we will continue to pay attention to this issue in our future research works.

Reference: Nguyen TA, Heng JWJ, Ng YT, Sun R, Fisher S, Oguz G, Kaewsapsak P, Xue S, Reversade B, Ramasamy A, Eisenberg E, Tan MH. Deep transcriptome profiling reveals limited conservation of A-to-I RNA editing in *Xenopus*. BMC Biology. 2023, 21(1):251.

(2) "Two miRNA-target prediction software programs, miRanda and RNAhybrid, were used to identify the miRNAs that potentially act on CYP18A1. The results showed that miR-3036-5p could bind to the sequence containing edited position (editing site 528) of CYP18A1 in M. dirhodum." Is there any other miRNA that can also act on CYP18A1, thereby regulating its expression?

The predicted results indicate that there are several other miRNAs can act on *CYP18A1*, but none of them can bind to this editing site (editing site 528). Therefore, we did not pay attention to other miRNAs.

(3) 11678 A-to-I RNA-editing sites were systematically identified in *M. dirhodum*. Does that mean RNAi-mediated knockdown of ADAR2 may affect the RNA-editing and expression of a large number of genes? Please clarify.

It is of course possible that RNAi-mediated knockdown of *ADAR2* may affect the RNA-editing and expression of a large number of genes. A-to-I RNA editing was also observed in 5 other genes that involved in 20E biosynthesis and signaling pathway, but no evident difference was identified for the RNA editing and expression levels of these 5 genes after crowding treatment (Fig. S2, Table S5). That means the A-to-I RNA editing of *CYP18A1* might be crucial in 20E-mediated wing dimorphism in *M. dirhodum*.

(4) It is interesting that "the transcriptional level of *ADAR2* was 2.19 fold higher in the crowding+20E treatment parent than that in the normal group, but no significant difference was identified between the crowding and normal groups". ADAR2 can be induced by 20E, rather than crowding. How should the author explain? It seems that 20E induction can also cause many RNA editing events.

20-hydroxyecdysone (20E) can affect the growth and development, molting, metamorphosis, and reproductive processes of insects. According to this result, 20E induction can also cause RNA editing events by regulating the expression of *ADAR2*, and which may provide valuable references for the future study on 20E. Meanwhile, we will also continue to pay attention to this issue in our future research works.

(5) Authors provided a lot of text to describe the genome assembly. I don't think it's necessary, authors can make appropriate deletions.

Thank you for this suggestion. This is the first high-quality chromosome-level genome of *M. dirhodum*, which will be very helpful for the cloning, functional verification, and evolutionary analysis of genes in this important species or even other Hemiptera insects. Therefore, I think it is necessary to provide a detailed description. We will also make appropriate deletions in the “Result and Discussion” sections.

**Reviewer #2 (Recommendations For The Authors):**
Additional concerns- With an existing genome sequence available for the peas aphid **Acyrthosiphon pisum**, why have these authors chosen to use the rose-grain aphid for this study? It would be helpful to address any limitations in **Acyrthosiphon pisum** or advantages in **Metopolophium dirhodum** that explain that decision.

Wheat is one of the main grain crops in China as well as in the world. *Metopolophium dirhodum* is one of the most important wheat aphids around China, and has posed a significant threat to grain production. The current study was conducted to determine the regulatory mechanism of wing dimorphism on *M. dirhodum*, which might be of great significance to better control this pest in wheat production.

Surely the pea aphid offers more established experimental tools and genomic resources. However, with the development of high-throughput sequencing technology, the chromosome level genomes of many insect species have been assembled. That means any of various insects might be studied as a model species, and not limited to *Drosophila melanogaster*, *Acyrthosiphon pisum*, etc.

- In Figure 5E, what anatomy is being shown in FISH? Moreover, this represents a single sample. It would be preferable to include a supplemental figure with comparable images from at least 3 additional specimens.

It is the whole aphid body, and we have already uploaded additional 2 FISH images to the supplementary material Fig. S5. Thank you for this suggestion.

- L190: Conservation alone seems inadequate to conclude that a chromosome functions as a sex chromosome. It would be fine to note the homology between Chr1 and the X of other Aphidini, but there are other explanations for that. Inference that Chr 1 is a sex chromosome might come from observations in karyotypes (by relative size comparisons or ideally from FISH) or from comparison of reads mapped to the chromosomes, suggesting Chr1 is hemizygous in males.

Karyotype analysis experiment was not conducted in this research, so here the sex chromosome was determined based on chromosome homology between *M. dirhodum* and *A. pisum* genome. We have made appropriate modifications to the description in the article. Thank you for this suggestion.

- L205: It's unclear to me how to interpret RNA editing results, based on RNAseq data, that map to "intergenic regions", especially when this is such a large fraction (37.3%) of the total result. Does this suggest a fundamental problem with the analysis, that so much RNAseq data maps to parts of the genome that are not annotated as genes?

Non-coding RNA regions often account for a large proportion in the genome, and this RNAseq data is mapped to non-coding RNA transcription regions (37.3%) between protein-coding genes (intergenic regions).

- L288-290: What degrees of confidence are attached to the predictions of these miRNA targets?

There is no clear research indicating the accuracy of miRNA target prediction software. However, by comprehensively utilizing multiple prediction tools and experimental verification, the accuracy and reliability of prediction can be significantly improved.

Actually, the prediction of miRNA targets is only a preliminary identification step, and we have subsequently demonstrated that miR-3036-5p can act on *CYP18A1* through dual-luciferase reporter assay, RNA immunoprecipitation and FISH, etc.

- L296-298: The mechanism proposed in this study seems to imply that miR-3036-5p should be absent (not expressed) in aphids under crowded conditions. Therefore, relative realtime PCR is not particularly useful here. Finding that the miR relative expression is reduced by 48.8% is meaningless, because in *relative* expression, zero has no special meaning. In this case, absolute quantitative PCR, measuring actual transcript numbers, would be far more informative.

miR-3036-5p is not absent in aphids under crowded conditions. Only a significant decrease of miR-3036-5p in expression level under crowded conditions was identified compared to normal feeding conditions (Fig. 5B). So it should be reasonable to use relative quantitative methods for expression level analysis.

- L361: Isn't alternative mRNA splicing a more common post-transcriptional modification?

I'm very sorry, this sentence has been modified to “A-to-I RNA editing is one of the most prevalent forms of posttranscriptional modification in animals, plants, and other organisms.” Thank you for this suggestion.

- L372: "Functional wing polymorphism is commonly observed in insects as a form of adaptation and a source of variation for natural selection (14)." The relationship between plastic phenotypic variation and natural selection is complex, and there is a large theoretical literature in evolutionary biology and evo-devo on this topic, but it is not a focus in the cited review by Zhang et al.. It would be helpful if the authors could expand on this idea with reference to some of this literature (e.g. Levins 1968; Harrison 1980; Moran 1992; Roff 1996; West-Eberhard 2003; Zera 2009).

I have changed the citation and expanded on this idea. “Wing polymorphism is commonly observed in insects, resulting from variation in both genetic factors and environmental factors (Zera 2009).”

- L404: Use the word "accurate" seems inappropriate in this context. Both morphs are equally "accurate".

This sentence has been modified to “resulting in the alteration of *CYP18A1* expression and wing dimorphism of offspring regulated by miR-3036-5p”, Thank you for this suggestion.

- L412: Reference 67 seems irrelevant to this point.

References have been changed and added.

67. E.J. Duncan, C.B. Cunningham, P.K. Dearden. Phenotypic plasticity: what has DNA methylation got to do with it? Insects. 13(2):110 (2022).

68. K.J. Rangan, S.L. Reck-Peterson, RNA recoding in cephalopods tailors microtubule motor protein function. Cell 186, 2531-2543 (2023).

- L443: Is this referring to "mixed stage" aphids?

Yes. To make it clearer, this sentence has been modified to “Approximately 200 mg of fresh *M. dirhodum* with mixed stages (including first- to fourth-instar nymphs and winged and wingless adults)”.

- L483: What mass or number of individual aphids was used? I assume multiple individuals were pooled?

Each sample contains approximately 200 aphids.

- L499: Why was k = 17 used? The default is k = 21.

The selection of k is usually an odd number between 15 and 21, which ensures that the types of k-mers can cover the genome while being small enough to avoid erroneous effects. Therefore, using 17 is very reasonable.

- L574: what does it mean "multiple editing types"? What different types are possible? Are you referring to things other than A-to-I editing?

That means besides A-to-I, this locus may also have other editing situations, such as A-to-C. If this situation occurs, it will be discarded.

- L635: Which luciferase construct or plasmid has been used in this experiment? Citation to that source is necessary.

PmirGLO vector (Promega, Leiden, Netherlands) was used in this experiment, and a reference has been added.

B. Zhu, L. Li, R. Wei, P. Liang, X. Gao. Regulation of GSTu1-mediated insecticide resistance in Plutella xylostella by miRNA and lncRNA. *PLoS Genetics*. 17(10), e1009888 (2021).

- L644: Did cDNA synthesis employ random primers or a poly-dT primer?

This kit provides mixed primers, including random and poly-dT primers. (PrimeScript RT reagent Kit with gDNA Eraser (Perfect Real Time), Takara Biotechnology, Dalian, China).

- Fig 4D: Seems like this panel should be divided to cover the two sites, as in Fig 3F. Right now the x-axis labels seem redundant.

Done. Thank you for this suggestion.

- Fig 7: Consider adding ADAR2 to this figure.

Done. Thank you for this suggestion.

- Table 1: It would be helpful to represent this data in a figure where the phylogenetic relationships among the species can be shown.

The phylogenetic relationships among the species were shown in Fig. 1D, and the table here may present genome information in more detail.